# Dynamic persistence of UPEC intracellular bacterial communities in a human bladder-chip model of urinary tract infection

Kunal Sharma, Neeraj Dhar[†]*, Vivek V Thacker[†]*, Thomas M Simonet, Francois Signorino-Gelo, Graham W Knott, John D McKinney*

School of Life Sciences, Swiss Federal Institute of Technology in Lausanne (EPFL), Lausanne, Switzerland

**Abstract** Uropathogenic *Escherichia coli* (UPEC) proliferate within superficial bladder umbrella cells to form intracellular bacterial communities (IBCs) during early stages of urinary tract infections. However, the dynamic responses of IBCs to host stresses and antibiotic therapy are difficult to assess in situ. We develop a human bladder-chip model wherein umbrella cells and bladder microvascular endothelial cells are co-cultured under flow in urine and nutritive media respectively, and bladder filling and voiding mimicked mechanically by application and release of linear strain. Using time-lapse microscopy, we show that rapid recruitment of neutrophils from the vascular channel to sites of infection leads to swarm and neutrophil extracellular trap formation but does not prevent IBC formation. Subsequently, we tracked bacterial growth dynamics in individual IBCs through two cycles of antibiotic administration interspersed with recovery periods which revealed that the elimination of bacteria within IBCs by the antibiotic was delayed, and in some instances, did not occur at all. During the recovery period, rapid proliferation in a significant fraction of IBCs reseeded new foci of infection through bacterial shedding and host cell exfoliation. These insights reinforce a dynamic role for IBCs as harbors of bacterial persistence, with significant consequences for non-compliance with antibiotic regimens.

*For correspondence:
neeraj.dhar@epfl.ch (ND);
vivekvthacker@gmail.com (VVT);
john.mckinney@epfl.ch (JDMK)

[†]These authors contributed equally to this work

Competing interests: The authors declare that no competing interests exist.

## Introduction

Urinary tract infections (UTIs), the second most-common cause for the prescription of antibiotics (*Foxman, 2010*) are characterized by the high frequency of recurrence, defined as a reappearance of infection within 12 months despite the apparently successful completion of antibiotic therapy. Recurrence occurs in about 25% of all UTIs (*Foxman et al., 2000*) and strongly impacts the cost of healthcare and reduces the quality of life, particularly since more than 60% of women are diagnosed with a UTI at least once in their lifetime (*Klein and Hultgren, 2020*). Uropathogenic *Escherichia coli* (UPEC), the causative agent for the majority of UTIs, exhibits a complex lifestyle in the bladder, with planktonic sub-populations within the urine co-existing with intracellular bacteria. UPEC invasion of the urinary bladder generates substantial changes to bladder morphology and a robust immune response. Much of our current understanding of early stages of UTI and the intracellular lifestyle is derived from studies in the mouse model (*Anderson, 2003*; *Duraiswamy et al., 2018*; *Hannan et al., 2012*; *Hung et al., 2009*; *Justice et al., 2004*; *Schwartz et al., 2011*; *Yang et al., 2019*). Examinations of mouse bladder explants via microscopy have revealed the formation of intra-cellular bacterial communities (IBCs) composed of thousands of bacteria within individual superficial bladder cells (*Justice et al., 2004*). IBCs also play a prominent role in clinical infection and have also been harvested from the urine of cystitis patients (*Robino et al., 2013*; *Rosen et al., 2007*). These structures are considered to have a linear progression from an early stage of colonization by single bacteria to an intermediate stage of biofilm-like communities culminating in the release of bacteria

**eLife digest** Urinary tract infections are one of the most common reasons people need antibiotics. These bacterial infections are typically caused by uropathogenic *Escherichia coli* (also known as UPEC), which either float freely in the urine and wash away when the bladder empties, or form communities inside cells that the bladder struggles to clear. It is possible that the bacteria living within cells are also more protected from the immune system and antibiotics. But this is hard to study in animal models.

To overcome this, Sharma et al. built a 'bladder-chip' which mimics the interface between the blood vessels and the tissue layers of the human bladder. Similar chip devices have also been made for other organs. However, until now, no such model had been developed for the bladder.

On the chip created by Sharma et al. is a layer of bladder cells which sit at the bottom of a channel filled with diluted human urine. These cells were infected with UPEC, and then imaged over time to see how the bacteria moved, interacted with the bladder cells, and aggregated together. Immune cells from human blood were then added to a vascular channel underneath the bladder tissue, which is coated with endothelial cells that normally line blood vessels. The immune cells rapidly crossed the endothelial barrier and entered the bladder tissue, and swarmed around sites of infection. In some instances, they released the contents of their cells to form net-like traps to catch the bacteria. But these traps failed to remove the bacteria living inside bladder cells.

Antibiotics were then added to the urine flowing over the bladder cells as well as the vascular channel, similar to how drugs would be delivered in live human tissue. Sharma et al. discovered that the antibiotics killed bacteria residing in bladder cells slower than bacteria floating freely in the urine. Furthermore, they found that bacteria living in tightly packed communities within bladder cells were more likely to survive treatment and go on to re-infect other parts of the tissue.

Antibiotic resistance is a pressing global challenge, and recurrent urinary tract infections are a significant contributor. The bladder-chip presented here could further our understanding of how these bacterial infections develop in vivo and how good antibiotics are at removing them. This could help researchers identify the best dosing and treatment strategies, as well as provide a platform for rapidly testing new antibiotic drugs and other therapies.

at the late stage (*Anderson, 2003*; *Justice et al., 2004*). However, long-term imaging of infected animals or of explant-tissue is technically challenging and therefore it has been difficult to capture the dynamic changes that underlie the formation of these structures and their impact on infection and clearance by subsequent antibiotic treatment.

In addition, the bladder is an extremely complex organ; it has a stratified architecture with well-differentiated cell types; a lumen filled with urine whose composition and chemical properties can vary depending on the physiological state of the individual and is subjected to periodic and large changes in organ volume and surface area (*Korkmaz and Rogg, 2007*). These features may play important roles in infection, but they have been hitherto hard to capture outside of a whole animal model, where they are present in their entirety and cannot be dissected in a modular manner.

Organotypic models are well-suited to address these outstanding questions. A key strength of advanced organotypic models such as organs-on-chip (*Benam et al., 2016*; *Huh et al., 2010*; *Jang et al., 2013*; *Jang et al., 2019*; *Kim et al., 2016*; *Novak et al., 2020*; *Zhou et al., 2016*) or organoids (*Cakir et al., 2019*; *Clevers, 2016*; *Rossi et al., 2018*; *Sato et al., 2009*) is that they increasingly recapitulate the complexity of host physiology. Organ-on-chip models have been developed for several organs either in isolation or in combination (*Novak et al., 2020*; *Ronaldson-Bouchard and Vunjak-Novakovic, 2018*). These systems are increasingly being used to model infectious diseases including viral and bacterial infections of the respiratory tract (*Nawroth et al., 2020*; *Si et al., 2021*; *Thacker et al., 2020*; *Thacker et al., 2021*), gut (*Jalili-Firoozinezhad et al., 2019*; *Kim et al., 2016*; *Shah et al., 2016*; *Tovaglieri et al., 2019*), kidney (*Wang et al., 2019*), and liver (*Kang et al., 2017*). Recently, bladder organoids that mimic the stratified architecture of the urothelium have been used to study infections (*Horsley et al., 2018*; *Smith et al., 2006*; *Sharma et al., 2021*). However, the organoid model suffers from certain shortcomings inherent to the 3-D architecture such as the lack of vasculature, the inability to manipulate the cells mechanically, and the

constrained volume of the lumen. In contrast, organ-on-chip models offer a complementary approach that does not suffer from these limitations. However, to our knowledge, there have been no reports of a bladder-on-chip model. Although there are studies that have developed in vitro bladder models that recreate the stratified architecture of the bladder epithelium (*Horsley et al., 2018*; *Smith et al., 2006*; *Suzuki et al., 2019*), they have not been used to recapitulate the multiple stages of IBC formation. Similarly, while a few studies have visualized different stages of IBC development by culturing human bladder cells under urine flow, these models are restricted to monoculture experiments (*Andersen et al., 2012*; *Iosifidis and Duggin, 2020*). Furthermore, the lack of vasculature in these models restricts the extent to which immune cell components or drugs can be introduced in a physiologically relevant manner and none of the systems reported to date offer the possibility of mimicking the mechanics of filling and voiding in a functioning bladder (*Andersen et al., 2012*; *Horsley et al., 2018*; *Iosifidis and Duggin, 2020*; *Smith et al., 2006*).

Here, we report the development and characterization of a bladder-chip model that mimics the bladder architecture by co-culture of a well-characterized human bladder epithelial cell line with bladder microvascular endothelial cells in a device geometry that allows the two cell types to be exposed to urine and nutritive cell culture media, respectively. The flow rates in the apical and vascular channels can be controlled independently, and multiple rounds of micturition are recreated via the application of a linear strain to the PDMS membrane that serves as the substrate for co-culture. Using this model, we show that diapedesis of neutrophils to sites of infection on the epithelial side can lead to the formation of neutrophil swarms and neutrophil extracellular traps (NETs), and that IBCs offer substantial protection to bacteria from antibiotic clearance. Our observations suggest that the role of IBCs in reseeding bladder infections upon cessation of antibiotic treatment or failure to complete a course of antibiotics might be more important than previously assumed and therefore strategies aimed toward eradication of IBCs are very crucial for treatment efficacy.

## Results

### Reconstitution of bladder uroepithelium and bladder vasculature

We established a bladder-chip infection model by co-culturing HMVEC-Bd primary human bladder microvascular endothelial cells with the 5637 human bladder epithelial cell line (HTB-9, epithelial cells) in a novel bladder-chip approach (*Figure 1A*). In a small subset of experiments, primary human bladder epithelial cells (primary epithelial cells) were used in place of 5637 cells. Immunostaining verified that the epithelial and endothelial cells formed confluent monolayers (cell densities in *Table 1*) with high expression of junction markers such as epithelial cell adhesion molecule (EpCAM) in the epithelial cell layer, and platelet endothelial cell adhesion molecule-1 (PECAM-1) and VE-cadherin in the endothelial cell layer (*Figure 1B,C*, *Figure 1—figure supplement 1A–H*). Expression of these and other junction markers such as E-cadherin and zonula occludens-1 (ZO-1) was consistent across monocultures (*Figure 1—figure supplement 2A–G*).

Similarly, a majority of the epithelial cells expressed cytokeratin 7 (CK7), a general uroepithelial marker and cytokeratin 8 (CK8), a differentiated uroepithelial marker both in monoculture (*Figure 1—figure supplement 2I,J*) and on-chip (*Figure 1B*, *Figure 1—figure supplement 1B*, *Figure 1—figure supplement 1F*). Some endothelial cells were also CK7+ (*Figure 1C*, *Figure 1—figure supplement 2H*). Co-culture of bladder epithelial and bladder endothelial cells in the human bladder-chip therefore did not alter the cellular expression patterns of these markers as compared to monocultures. We further characterized the 5637 cells in monoculture for markers for uroepithelial differentiation. A total of 5637 cells showed high expression of uroplakin-3a (UP3a) which has been shown to be essential for UPEC infection (*Figure 1—figure supplement 2K*; *Martinez et al., 2000*; *Mulvey et al., 1998*) but only a small proportion of epithelial cells were positive for the basal cell marker cytokeratin 1 (CK1), in agreement with the umbrella cell nature of the 5637 cells (*Figure 1—figure supplement 2L,M*; *Duncan et al., 2004*; *Smith et al., 2006*). Overall, these observations confirmed that the bladder-chip is populated with cells that mimic the physiology of the bladder vasculature and the superficial uroepithelial cell layer, although the relatively small size of the epithelial cells is likely an artefact inherent in the 5637 cells. Further, the chip design enables the flow of media with different compositions through the epithelial and vascular channels. We used this feature to

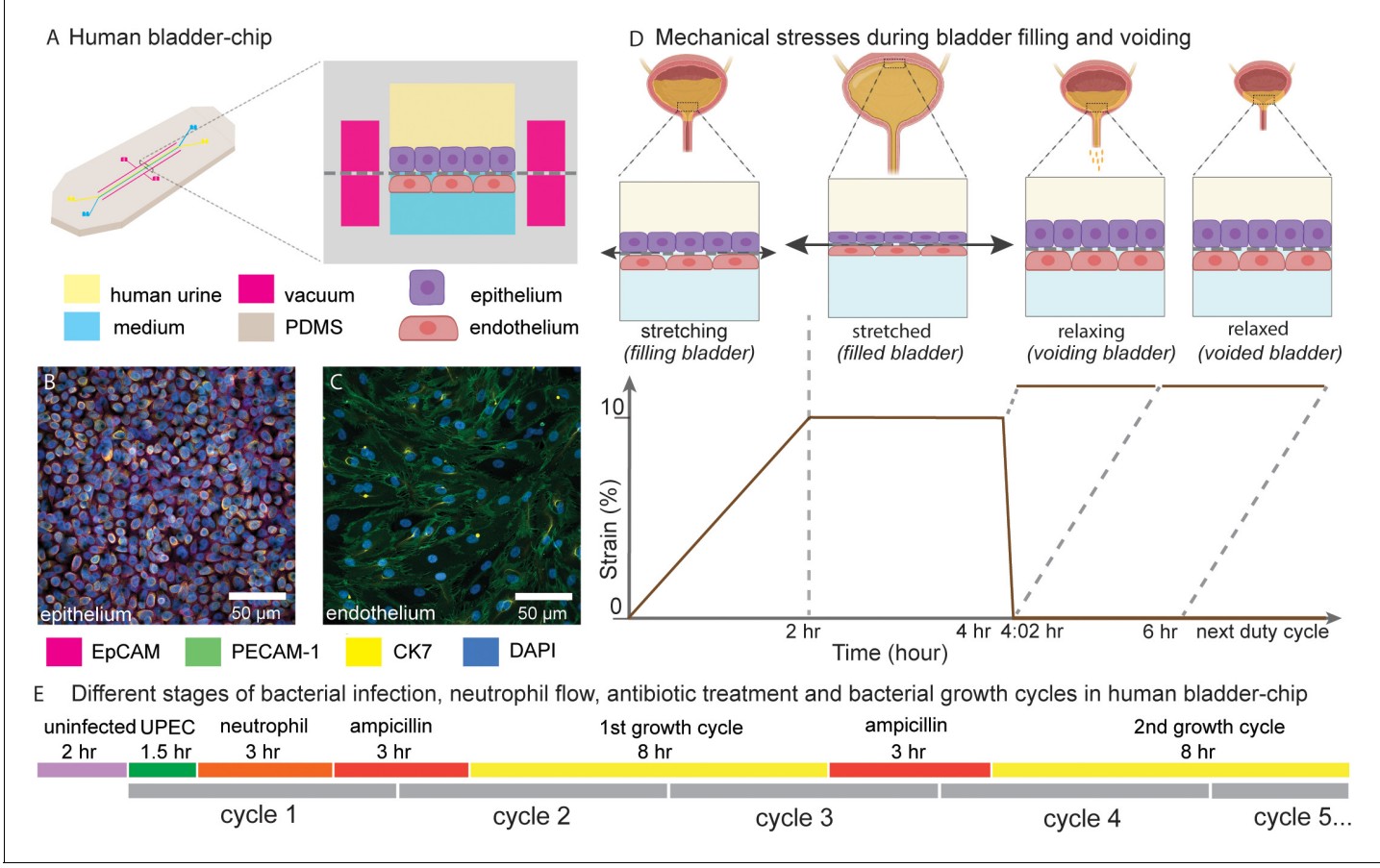

**Figure 1.** Human Bladder-chip model of UTI recapitulates the physiology of bladder filling and voiding. (**A**) Schematic of the human bladder-chip with co-culture of the 5637 human bladder epithelial cell line (epithelium, top) and primary human bladder microvascular endothelial cells (endothelial, bottom) on either side of the stretchable and porous membrane. Pooled human urine diluted in PBS and endothelial cell medium were perfused in the apical and vascular channels respectively to mimic bladder physiology. A negative pressure in the 'vacuum' channels (magenta) on either side of the main channel was applied to stretch the porous membrane to mimic stretching of the bladder. (**B, C**) Immunofluorescence staining of confluent epithelial and endothelial cell monolayers (anti-EpCAM (magenta) and anti-CK7 (yellow) for the epithelial cells and anti-PECAM-1 (green) for the endothelial cells) in an uninfected control chip. Some endothelial cells also stained positive for CK7. Cell nuclei were labeled with DAPI (azure). (**D**) Schematic of the reconstitution of the bladder filling and voiding cycle via stretching of the membrane with a duty cycle of 6 hr. The cycle consisted of a linear increase in strain through stretching of the membrane (*filling bladder*, 0–2 hr), maintenance of the membrane under stretch (*filled bladder*, 2–4 hr), a quick relaxation of applied strain over 2 min (*voiding bladder*, 4:02 hr) and maintenance without applied strain (*voided bladder*, 4:02 hr to 6 hr). (**E**) An overview of the timeline of the experimental protocol including infection, addition of neutrophils via the vascular channel, and two cycles of antibiotic treatment interspersed by two bacterial growth cycles. The consecutive bladder duty cycles are indicated.

The online version of this article includes the following figure supplement(s) for figure 1:

**Figure supplement 1.** Characterization of co-cultures of bladder epithelial cells and bladder endothelial cells in bladder-chip.

**Figure supplement 2.** Characterization of monocultures of 5637 bladder epithelial cells and HMVEC-Bd bladder microvascular endothelial cells.

**Figure supplement 3.** Quantification of the linear strain in the PDMS membrane as a function of applied negative pressure in the vacuum channels of the bladder-chip.

**Table 1.** Characterization of epithelial and endothelial cell densities from a total of n=18 fields of view in both the epithelial and endothelial layers from n=two bladder-chips.

| Cell type | Cell density/$10^4$ $\mu m^2$ |
| --- | --- |
| Epithelial | 31.7 ± 5.1 |
| Endothelial | 4.0 ± 0.6 |

mimic bladder physiology by perfusion of diluted pooled human urine on the epithelial side and endothelial cell medium on the vascular side.

## Modeling bladder filling and bladder voiding in the bladder-chip

The human bladder experiences vast changes in volume and surface area on a periodic basis. The first phase of this cycle, ('*filling bladder* state'), occurs through a slow and gradual increase in bladder volume due to addition of urine from kidney via the ureters, resulting in a large increase in bladder volume (*Figure 1D*). At the level of the uroepithelium, these changes manifest as a two-dimensional biaxial stretch of the bladder tissue. Ex vivo studies in rat bladders report that a maximum strain of 90% and 130% can be applied to bladder explants in the circumferential and longitudinal directions (*Gloeckner et al., 2002*; *Parekh et al., 2010*), although micturition in vivo is likely triggered before this maximum value is reached. Subsequently, volume growth slows, and the bladder attains a relatively constant volume, with the bladder epithelium in a corresponding state of maximal stretch, ('*filled bladder* state'). Voiding of the bladder through urination rapidly reduces bladder volume, ('*voiding bladder* state') and removes the strain experienced by the bladder cells. The bladder epithelium, as a viscoelastic material, responds to the removal of strain and relaxes over the subsequent period (*De Pascalis et al., 2018*) where the bladder volume remains low ('*voided bladder* state') (*Figure 1D*).

The architecture of the bladder-chip device allows for a linear strain to be applied to the porous membrane via negative pressure in the channels adjacent to the main channel of the device ('the vacuum channels') (*Grassart et al., 2019*; *Huh et al., 2010*; *Figure 1A*). We characterized the linear strain experienced by epithelial cells due to application of negative pressure (*Figure 1—figure supplement 3*). A plot of applied pressure vs. strain was linear, and we achieved a dynamic range of linear strain (0% to 19%) by application of negative pressure (0 to −900 mbar). High levels of applied strain were incompatible with long-term time-lapse microscopy as it generated a significant drift in the axial position of the membrane. We therefore limited the linear strain applied to a maximum of 10%, which is of the same order of magnitude, albeit a small fraction of the typical strain experienced by the bladder tissue in vivo. We then modeled the bladder filling and voiding over a 6 hr duty cycle with different states: *filling bladder* (0 to 2 hr, 0% to 10% strain), *filled bladder* (2 to 4 hr, 10% strain), *voiding bladder* over a period of 2 min (4 hr to 4:02 hr, 10% to 0% strain), and *voided bladder* (4:02 hr to 6 hr, 0% strain) (*Figure 1D*). This 6 hr duty cycle was repeated for the remaining duration of each experiment. Overall, the bladder-chip model enables co-culture of two cell-types in nutritionally different microenvironments with an applied strain that partly mimics the physiology of bladder filling and voiding cycles.

## UPEC infection of the epithelial layer under flow in the bladder-chip model

UPEC attachment and invasion of bladder epithelial cells has been shown to be sensitive to fluidic shear stress (*Andersen et al., 2012*; *Zalewska-Piątek et al., 2020*). Although UPEC infection in the bladder can occur in the presence or absence of shear stress, we infected the epithelial layer on the bladder-chip under a flow rate of 1.2 ml/hr (corresponding to a shear stress of 0.02 dyne cm$^{-2}$) for a period of 1.5–2 hr. Low shear stress conditions have been recently reported to enhance bacterial adhesion to epithelial cells (*Zalewska-Piątek et al., 2020*) and likely reflects the in vivo environment except during urination when the shear stress increases dramatically to 3–5 dyne cm$^{-2}$ (*Sokurenko et al., 2008*). This choice was also motivated by the small volumes of the microfluidic chip, as rapid bacterial growth in the diluted human urine could lead to acidification of the medium. Under these conditions, an overwhelming majority of bacteria did not attach to the epithelial cells. Time-lapse microscopy showed that bacterial attachment to the epithelial cells increased steadily over this period, but that the typical infectious dose at the end of this period was low (<one bacterium per epithelial cell, *Figure 2—figure supplement 1*).

Accordingly, we established a 28 hr live-cell imaging experimental protocol as shown in the schematic in *Figure 1E* and described in greater detail in the subsequent sections and in the Materials and methods. Briefly, this consisted of an initial period of acclimatization, followed by UPEC infection, introduction of neutrophils, and two consecutive cycles of antibiotic treatment and recovery to monitor IBC dynamics at the single-cell level. This experimental protocol therefore mimicked key

aspects of the host-pathogen interactions in early stages of UTIs as well as the response to antibiotic treatment.

## Diapedesis of neutrophils across the epithelial-endothelial barrier in response to UPEC infection in the bladder-chip

The bladder-chip platform enables continuous imaging while maintaining flow in both epithelial and vascular channels and mechanically stretching the membrane as part of the bladder voiding cycle. At the start of each live-cell imaging experiment, bladder epithelial cells in an initial relaxed state were perfused with sterile pooled human urine diluted in PBS on the epithelial side and endothelial cell medium on the vascular side respectively (*Figure 2A1, B1*, *Figure 2—figure supplement 2A1*). The epithelial side was inoculated with a low dose of UPEC in diluted urine and infection was performed under flow (*Figure 2A2, B2*, *Figure 2—figure supplement 2A2*) for ca. 1.5–2 hr. The first bladder duty cycle was also initiated at this timepoint. We maintained the optical focus of the microscope on the epithelial layer over the subsequent course of infection. UPEC attachment was evident during this infection phase by visual inspection (*Figure 2B2*, *Figure 2—figure supplement 2A2*).

After this period of infection, unattached bacteria in the epithelial channel were removed by perfusion of diluted urine. As our aim was to study the interaction of UPEC with host cells, this perfusion was maintained throughout the experiment. Continuous perfusion reduced the accumulation of bacteria in the urine and enabled imaging of intracellular bacteria without large amounts of background noise in the fluorescence channels from planktonic bacteria. We subsequently performed a series of interventions that mimic the course of UTI infections and used time-lapse imaging to monitor the simultaneous changes in host-pathogen interaction dynamics. Immune cells were introduced into the vascular channel to mimic the host response to infection. Among innate immune cells, neutrophils are the first line of defense in UTIs (*Haraoka et al., 1999*). Neutrophils were isolated from human blood with a high degree of purity (*Son et al., 2017*) verified by immunostaining for the neutrophil-specific marker CD15 (*Zahler et al., 1997*; *Figure 2—figure supplement 3*).

Neutrophils were pre-labeled with the cytoplasmic CellTracker Deep Red dye to enable identification and tracking during time-lapse imaging and subsequently introduced in the bladder-chip devices through the vascular channel at a cell concentration of ca. 2 million cells/ml similar to the neutrophil concentration in human blood (*Hsieh et al., 2007*). This mimics the natural route of immune cell migration into the bladder (schematic in *Figure 2A3–A5*, snapshots in *Figure 2B3–B5*, *Figure 2—figure supplement 2A3-A5*) During this period, the flow rate through the vascular channel was increased to 3 ml/hr (shear stress $\eta$ = 1.0 dyne/cm$^2$), which aids neutrophil attachment to endothelial cells (*Alon et al., 1995*). In uninfected control chips, neutrophil attachment to the endothelial layer was minimal (*Figure 2—figure supplement 4A,B*) and diapedesis of neutrophils to the epithelial layer was rare (*Figure 2C*, *Figure 2—figure supplement 5A*). In stark contrast, infection of the epithelial layer with UPEC elicited a robust attachment of neutrophils to endothelial cells (*Figure 2—figure supplement 4C,D*), along with rapid diapedesis across to the epithelial side (*Figure 2B2–B4*, *Figure 2—figure supplement 2A2-A4*, *Figure 2—figure supplement 5C*). We confirmed that neutrophil diapedesis was also stimulated in the uninfected chips upon exposure to a gradient across the epithelial-endothelial barrier of pro-inflammatory cytokines known to be upregulated during UTIs - IL-1$\alpha$, IL-1$\beta$, IL-6, and IL-8, each at 100 ng/ml (*Agace et al., 1993*; *Hedges et al., 1992*; *Nagamatsu et al., 2015*; *Song et al., 2007*; *Figure 2—figure supplement 5B*). However, infection elicited a more robust neutrophil diapedesis response compared to the cytokine gradient, evident in the higher number of neutrophils observed across multiple fields of view on the epithelial side (*Figure 2—figure supplement 5B–D*). These results suggest that infection on-chip generates a strongly pro-inflammatory environment locally and that neutrophil diapedesis is further stimulated by the presence of UPEC (*Agace et al., 1995*; *de Oliveira et al., 2016*).

Using time-lapse imaging, we quantified the kinetics of neutrophil migration with a temporal resolution of up to 7.5 min. Neutrophil diapedesis to the epithelial sites of infection was detected as early as ca. 15–17 min post-introduction into the vascular channel (*Figure 2B3*, *Figure 2—figure supplement 2A3*, *Figure 2—video 1*, *Figure 2—video 2*). Neutrophils that migrated to the epithelial side aggregated on the epithelial cells (*Figure 2B4–B5*, *Figure 2—figure supplement 2A4-A5*, *Figure 2—video 1*, *Figure 2—video 2*) In some cases, these neutrophils were able to control bacterial growth (*Figure 2B4–B5*, solid yellow boxes), whereas in others, bacterial growth was uncontrolled (*Figure 2B4–B5*, dashed white boxes). Diapedesis was observed in every field of view

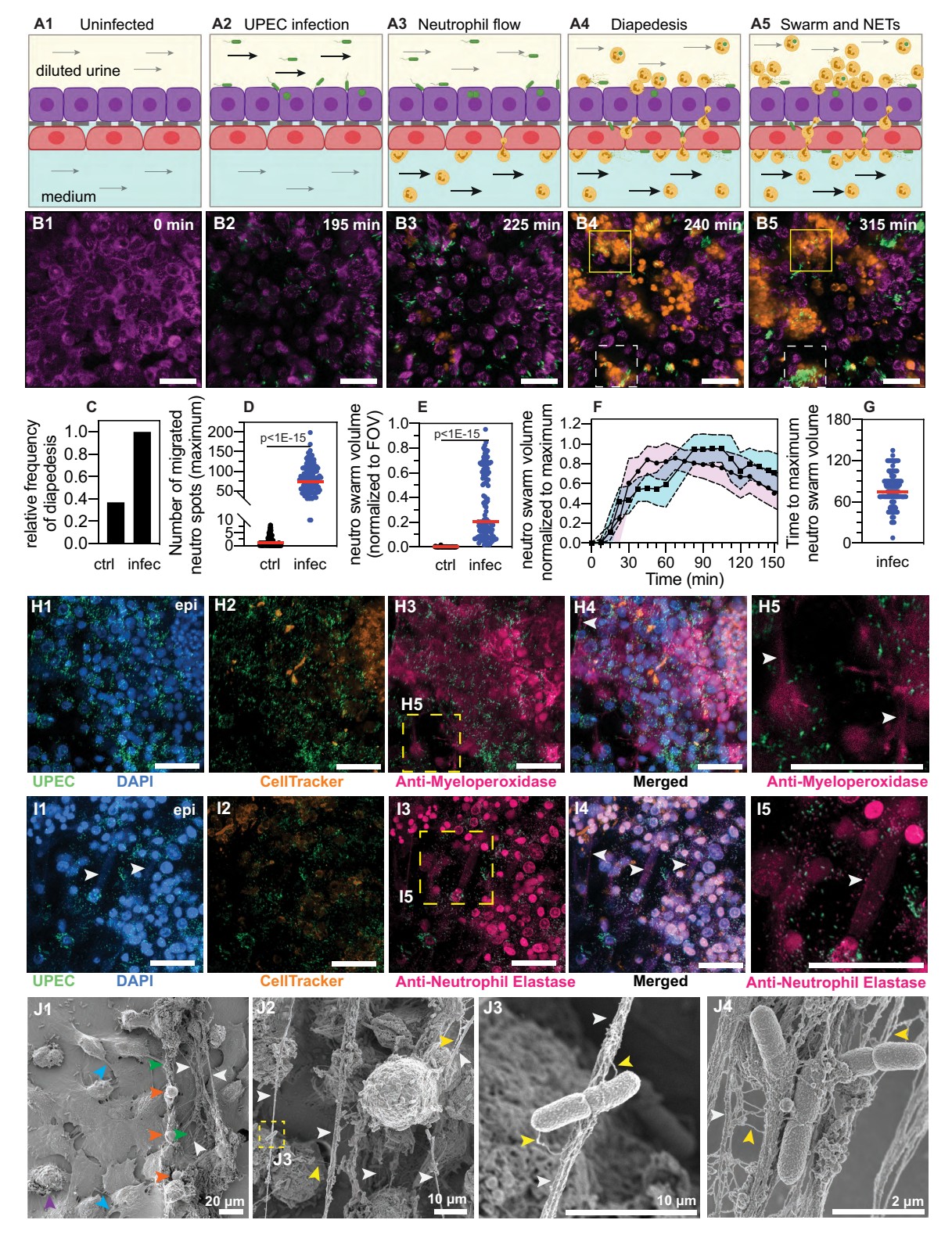

**Figure 2.** Neutrophil diapedesis and NET formation on-chip. (A1–A5) Schematic of the UPEC infection, introduction of neutrophils and diapedesis of neutrophils across the epithelial-endothelial barrier to sites of infection. Flow in the epithelial and endothelial channels is indicated by arrows; the flow rate was increased upon introduction of neutrophils in the vascular channel to increase attachment. Increased flow rate in the epithelial and endothelial channel is indicated by black arrows. (B1–B5) Snapshots from time-lapse imaging highlighting each stage in the infection cycle shown in (A1–A5).

*Figure 2 continued on next page*

*Figure 2 continued*

Bladder epithelial cells (magenta) and neutrophils (amber) were identified with membrane (Cell Mask Orange) and cytoplasmic (Cell Tracker Deep Red) dyes, respectively. UPEC identified via constitutive expression of YFP are shown in green. Neutrophil swarms could either control bacterial growth (yellow dashed square, compare B4 vs. B5) or did not manage to restrict bacterial growth (white dashed square, compare B4 vs. B5). (C) Bar charts for relative frequency of neutrophil diapedesis (black) in n=three uninfected control bladder-chips and n=four infected bladder-chips. Data obtained from n=95 and n=116 fields of view, each of which was 206 x 206 µm². (D) Quantification of the number of neutrophils detected on the epithelial layer, in control and infected bladder-chips. The red bar represents the median value, p<1E-15. In many instances in the uninfected control bladder-chips, no neutrophil diapedesis is detected. Data obtained from n=95 and n=130 fields of view, each of which was 206 x 206 µm². (E) A plot of the maximum neutrophil swarm volume on the epithelial layer normalized to the total volume for n=67 and n=118 fields of view on n=three uninfected control and n=three infected bladder-chips. The red bar represents the median value. p<1E-15. (F) A plot of neutrophil swarm volume on the epithelial layer over time for n=51 and n=40 fields of view for n=two technical replicates indicated by squares and circles. For each time profile, the volume is normalized to the maximum volume attained over the timeseries and t=0 refers to the timepoint at which neutrophils are introduced into the vascular channel (G) Plot of the time to reach the maximum swarm volume in n=154 fields of view across n=four infected bladder-chips. (H1–H5, I1–I5) NET formation by neutrophils on the epithelial layer. The neutrophils are identified by a cytoplasmic dye (CellTracker Deep Red, false colored in amber) (H2, I2) and immunostaining with an anti-myeloperoxidase antibody (H3, zooms in H5) or an anti-neutrophil elastase antibody (I3, zooms in I5). Merged images in each case are shown in H4 and I4. UPEC identified via YFP expression is shown in spring green. Nuclear labelling with DAPI is shown in azure. (H5) NETs, identified via anti-myeloperoxidase staining or anti-elastase staining are indicated with white arrows in (H5) and (I5), respectively. (J1–J4) Scanning electron micrographs of the epithelial layer of an infected bladder-chip 2 hr after the introduction of neutrophils in the endothelial channel. (J1) Neutrophils (amber arrowheads) are visible above a layer of epithelial cells. Thick bundles consisting of many thinner NET structures between neutrophils are indicated by white arrowheads, and examples of individual UPEC bacteria within the NETs are indicated by green arrowheads. A heavily infected epithelial cell with UPEC visible on the epithelial cell (purple arrowhead), and appendages between epithelial cells (cyan arrowheads) are also visible. (J2) Micrographs showing thick bundles consisting of many thinner NET structures (white arrowheads) that extend between cells and trap many individual bacteria. Thinner NET fibers (yellow arrowheads) are also visible. (J3) Zooms of the regions in J2 identified by a yellow dashed square. Two bacteria held by a thick bundle composed of many thinner NET fibers are shown. (J4) Micrograph highlighting multiple bacteria trapped between NET bundles. p-Values in D and E were calculated using a Mann-Whitney test. Scale bar = 50 µm in B1–B5, H1-H5, I1-I5.

The online version of this article includes the following video and figure supplement(s) for figure 2:

**Figure supplement 1.** Quantification of UPEC attachment to bladder epithelial cells on-chip under flow.
**Figure supplement 2.** Timeseries highlighting neutrophil diapedesis and swarm formation.
**Figure supplement 3.** Neutrophils isolated from human blood are CD15+.
**Figure supplement 4.** Neutrophil attachment to endothelial cells is enhanced upon bacterial infection.
**Figure supplement 5.** Neutrophil diapedesis is stimulated by a pro-inflammatory cytokine gradient across the epithelial-endothelial barrier.
**Figure supplement 6.** NETs formation on the epithelial and endothelial layers of an infected bladder-chip.
**Figure supplement 7.** SEM characterization of uninfected and infected bladder-chips.
**Figure supplement 8.** Neutrophils do not form NETs in response to shear stress in the bladder-chip.
**Figure 2—video 1.** Description: Infection of bladder-chip with UPEC, diapedesis of neutrophils across the epithelial-endothelial barrier and formation of swarms (*Figure 2B1–B5*).
https://elifesciences.org/articles/66481#fig2video1
**Figure 2—video 2.** Description: Neutrophil diapedesis and swarm formation on the epithelial side of UPEC infection (*Figure 2—figure supplement 2A1-A5*).
https://elifesciences.org/articles/66481#fig2video2

---

examined across infected chips (n=4), whereas diapedesis occurred infrequently in uninfected controls (*Figure 2C*). Furthermore, the size of these aggregates was significantly larger in infected bladder-chips vs. uninfected controls, with the large aggregates in infected-chips exhibiting the characteristic features of neutrophil swarms (*Kienle and Lämmermann, 2016*; *Kreisel et al., 2010*; *Lämmermann et al., 2013*). We quantified the size of the swarm-like aggregates by either estimating the maximum number of neutrophils observed in fields of view on the epithelial side (*Figure 2D*) or by measuring the total volume occupied by neutrophils normalized to the volume of a 3D field of view (*Figure 2E*). Using either metric, infection generated swarms that did not occur in uninfected bladder-chips. Time-lapse imaging confirmed that swarm formation was rapid (*Figure 2F*), and swarms reached their maximum size typically between 60- and 90-min post-introduction of neutrophils (*Figure 2G*).

## Formation of neutrophil extracellular traps (NETs) in response to UPEC infection

In addition to direct killing of engulfed bacteria, neutrophils, can also form neutrophil extracellular traps (NETs) through the release of cytosolic azurophilic granules around a scaffold of decondensed

chromatin (*de Oliveira et al., 2016*). These structures can trap and kill extracellular bacteria (*Brinkmann et al., 2004*). NETs consist of antimicrobial granules such as myeloperoxidase (*Metzler et al., 2011*) and neutrophil elastase (*Papayannopoulos et al., 2010*) and have been observed in urine harvested from UTI patients (*Yu et al., 2017*). Interestingly, infected bladder-chips immunostained at 2 hr post-introduction of neutrophils (3.5 hr post infection) with antibodies against neutrophil myeloperoxidase (*Figure 2H*, *Figure 2—figure supplement 6A1-A3*) and neutrophil elastase (*Figure 2I*, *Figure 2—figure supplement 6B1-B3*) showed areas on the epithelial layer which stained positive for each marker (*Figure 2H3 and I3*). In each example, zooms clearly show long filament-like structures that extend between cells and are either myeloperoxidase positive (*Figure 2H5*) or neutrophil elastase positive (*Figure 2I5*), morphological features that are strongly suggestive of NETs (*Brinkmann et al., 2004*; *Metzler et al., 2011*; *Papayannopoulos et al., 2010*). These areas had high numbers of neutrophils – a characteristic of swarms, and many of these neutrophils did not retain strong expression of the cytoplasmic CellTracker dye (*Figure 2H2, H3, I2 and I3*), suggestive of metabolic activation. Formation of NET-like structures was also observed in some instances by neutrophils in some locations on the endothelial channel, although large swarms did not form here (*Figure 2—figure supplement 6B1-B3*).

To further characterize these structures, we imaged the epithelial layer of uninfected and infected bladder-chips with or without the addition of neutrophils using scanning electron microscopy (SEM). Representative images of the epithelial layers of uninfected bladder-chip controls (*Figure 2—figure supplement 7A*) contrast with those from infected bladder-chips after neutrophil diapedesis (*Figure 2J1–J4*, *Figure 2—figure supplement 7C*); neutrophils on the surface of the epithelial layer are clearly distinguishable by their distinctive spherical morphology and size (*Figure 2J1*, amber arrowheads). Thick bundles consisting of many thinner fibers formed between these neutrophils are shown by white arrowheads and appear to extend between adjacent neutrophils (*Figure 2—figure supplement 7C*). Importantly, these structures did not form in infected bladder-chips without the introduction of neutrophils (*Figure 2—figure supplement 7B*), confirming that the NET-like architectures identified via immunofluorescence correspond to structures identified by SEM that morphologically resemble NETs (*Figure 2H,I*). Hereafter for simplicity, we shall refer to these structures as NETs. A series of higher magnification images in *Figure 2J2–J4* show the structure of the NETs in greater detail. Many individual UPEC bacteria (green arrowheads) are captured in these structures (*Figure 2J2–J4*), which are themselves composed of many individual thinner fibers (yellow arrowheads *Figure 2J2–J4*). In these images, a majority of the epithelial cells have a characteristic flattened morphology with cell appendages that can extend between cells (cyan arrowheads); this contrasts with the highly infected epithelial cell that is spherical in shape and with bacteria attached to its surface (*Figure 2J1*, purple arrowhead). Notably, we did not observe uroplakin plaques reported in vivo (*Kong et al., 2004*) on the surface of uninfected or heavily infected epithelial cells in the bladder-chip model.

Lastly, to verify that NET formation was a direct response to UPEC and not caused by hyperactivation of neutrophils due to experimental handling, we collected the neutrophils that passed through an infected bladder-chip without attachment. Even upon infection, an overwhelming majority of the cells perfused into the chip flow out without adhering, because of the geometry of the channels in the device. NET formation in these neutrophils was not observed unless the neutrophils were themselves infected by UPEC shed from the bladder-chip (*Figure 2—figure supplement 8*). The bladder-chip model is therefore able to recapitulate key aspects of the host response to the early stages of UPEC infection while demonstrating that intracellular bacteria were substantially protected from neutrophil-mediated killing.

## Heterogenous dynamics of intracellular bacterial communities within uroepithelial cells

Urinary tract infections that do not resolve upon intervention by the host immune system often require treatment with antibiotics. Antibiotics have complex pharmacokinetic and pharmacodynamic profiles in vivo, which we attempted to model through two successive rounds of a high dose administration of ampicillin - an antibiotic formerly prescribed for uncomplicated UTIs (ca. 40x MIC, *Figure 3—figure supplement 1A*), on both the epithelial and vascular side, interrupted by periods with no antibiotic (schematics in *Figures 3A* and *1E*). Continuous time-lapse imaging over this entire period allowed us to capture the responses of bacteria to this simplified model of antibiotic profiles

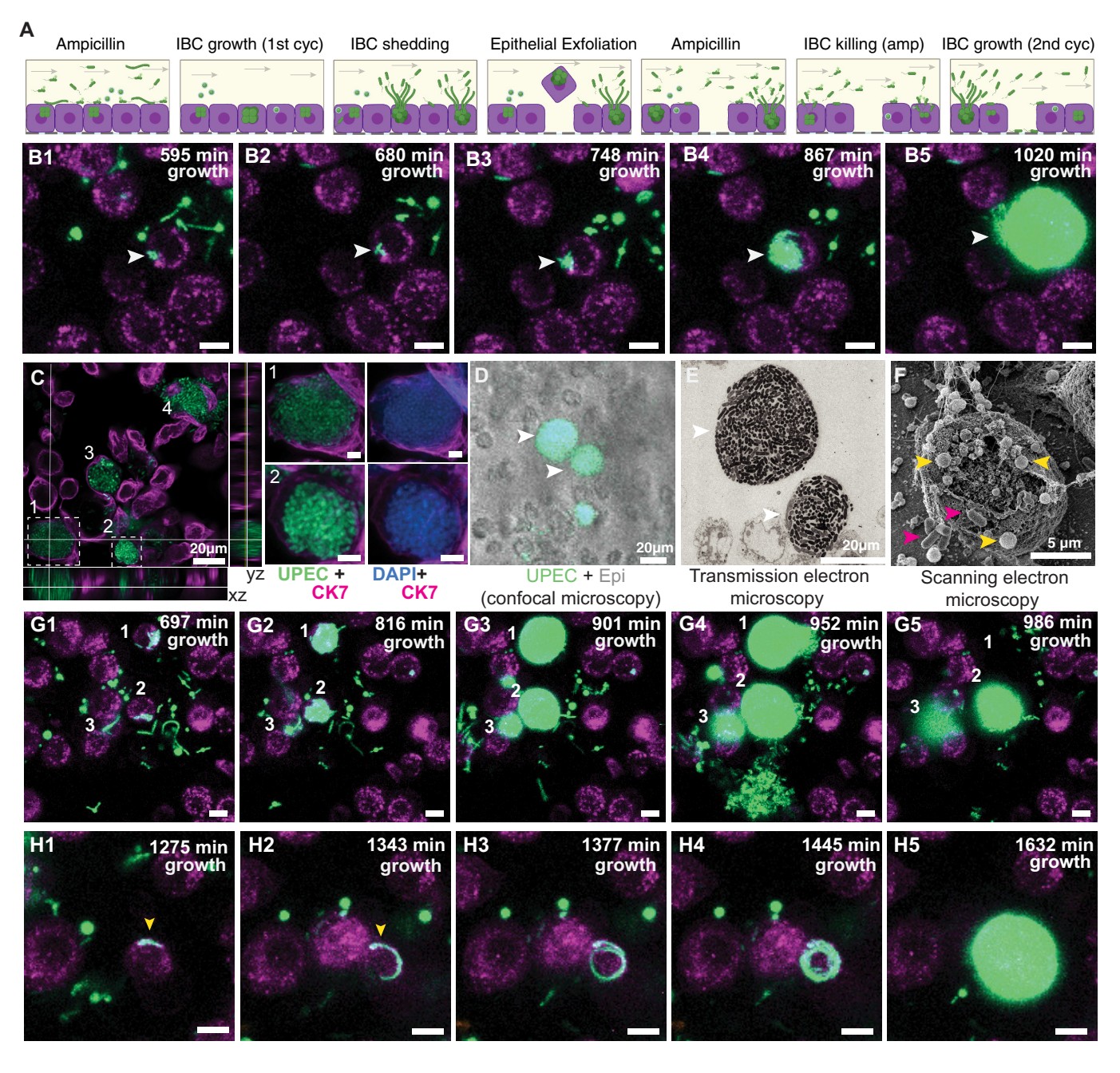

**Figure 3.** Bladder-chip reveals dynamics of IBC growth, shedding and exfoliation. (**A**) Schematics of the host-pathogen interactions within IBCs between successive rounds of antibiotic treatment with outcomes including shedding of bacteria and cell exfoliation shown. Arrows indicate the flow of diluted urine in the epithelial channel. (**B1–B5**) Timeseries for the growth of an IBC from few bacteria (white arrowhead) after the first round of ampicillin treatment. Bacteria grow to completely fill the cell volume (**B5**). (**C**) Immunofluorescence characterization of two IBCs; the intracellular nature of growth is confirmed with staining with an anti-CK7 antibody. Correlative light (**D**) and transmission electron micrographs (**E**) show two IBCs filled with both rod-shaped and coccoid-shaped bacteria. (**F**) Coccoid-shaped bacteria (yellow arrowheads) and rod-shaped bacteria (magenta arrowheads) are also visible in a scanning electron micrograph of an infected epithelial cell. IBCs on infected bladder-chips shown in **C–F** were fixed ca. 13.5 hr after UPEC infection and 6 hr into the first growth cycle. (**G1–G5**) Three examples of IBC growth, labeled 1–3 with differing outcomes. Unrestricted bacterial growth is observed within all IBCs. In IBC#1 at timepoint (**G4**) the IBC begins to shed bacteria into the surrounding medium. The cell exfoliates in the time interval between (**G4 and G5**). In IBC#2, bacterial shedding is visible at (**G4**), and shedding continues until end of the timeseries with a reduction in bacterial volume. In contrast, growth within IBC#3 is slower and neither shedding nor exfoliation occurs within the timeseries. (**H1–H5**) Timeseries highlighting an example of filamentous UPEC growth within an IBC. Scale bars, 10 µm in **B1–B5**, **G1–G5** and **H1–H5**.

*Figure 3 continued on next page*

*Figure 3 continued*

The online version of this article includes the following video and figure supplement(s) for figure 3:

**Figure supplement 1.** Non-growing UPEC in response to ampicillin administration in the bladder-chip.

**Figure supplement 2.** Immunofluorescence characterization of IBC formation.

**Figure supplement 3.** IBC formation from filamentous UPEC.

**Figure 3—video 1.** Formation of intracellular bacterial community inside epithelial cell arising from few bacteria (*Figure 3B1–B5*).

https://elifesciences.org/articles/66481#fig3video1

**Figure 3—video 2.** IBC shedding and exfoliation (*Figure 3G1–G5*).

https://elifesciences.org/articles/66481#fig3video2

**Figure 3—video 3.** Filamentous bacterial growth within an IBC (*Figure 3H1–H5*).

https://elifesciences.org/articles/66481#fig3video3

**Figure 3—video 4.** Filamentous bacterial growth within an IBC (*Figure 3—figure supplement 3A1-A5*).

https://elifesciences.org/articles/66481#fig3video4

and to study the persistence of UPEC upon antibiotic treatment in the different physiological niches (*Figure 3* and *Figure 4*).

A consequence of the first cycle of antibiotic administration under flow was the elimination of planktonic bacterial growth in the media in the apical channel, as is observed within the lumen of the bladder in vivo. Many bacteria that were still intact at the end of the antibiotic treatment intact did not regrow after the antibiotic was removed (*Figure 3—figure supplement 1B*). Regrowth commenced in only a small fraction of bacteria (*Figure 3—figure supplement 1C*) after a variable lag phase (*Figure 3—figure supplement 1D*) and rapid regrowth was highly correlated with intracellular location of bacteria within epithelial cells. These bacteria subsequently grew with a variable lag period to form large intracellular structures. These structures strongly resembled intracellular bacterial communities (IBCs) reported in vivo. An example of an early stage of the formation of IBC-like structures is shown in the timeseries in *Figure 3B1–B5*, *Figure 3—video 1*. This IBC was initially seeded with few bacteria that are present in *Figure 3B1* and clearly visible in *Figure 3B2*. At the end of the timeseries, the entire host cell was packed with bacteria (*Figure 3B5*). The exponential increase in bacterial numbers frequently led to the saturation of the 8-bit images in the bacterial channel. We also observed similar structures in infected chips reconstituted either with 5637 epithelial cells (*Figure 3—figure supplement 1E,F*) or primary epithelial cells (*Figure 3—figure supplement 1G–I*) and treated with ca. 8X MIC of fosfomycin (*Fedrigo et al., 2017*), a current frontline drug prescribed for UTIs with a different mode of action to ampicillin. This verified that the formation of these structures was independent of the host cell type and antibiotic used. Hereafter, we shall refer to these IBC-like structures as IBCs for simplicity, acknowledging that these structures may differ from IBCs reported in infected animals.

We therefore examined these structures in separate bladder-chips via immunofluorescence imaging with a higher spatial resolution (*Figure 3C*, *Figure 3—figure supplement 2*). Zooms of two of the four IBCs highlighted in *Figure 3C* show numerous tightly packed bacteria within epithelial cells that are both CK7+ (*Figure 3C*) and CK8+ (*Figure 3—figure supplement 2*). These bacterial morphologies are highly similar to early-stage IBCs observed in the bladder (*Anderson, 2003*; *Duraiswamy et al., 2018*; *Justice et al., 2004*). Higher resolution images of the biofilm-like structures within IBCs obtained via correlative light (*Figure 3D*) and transmission electron microscopy (*Figure 3E*) showed that bacteria in IBCs can be either coccoid or rod shaped, which is also evident in scanning electron micrographs of bacteria within IBCs (*Figure 3F*). IBCs within bladder-chip model therefore show many of the distinctive morphological features reported from images of infected bladders in the mouse model (*Anderson, 2003*; *Hunstad and Justice, 2010*).

Numerous reports from the mouse model of infection have shown IBCs to be dynamic structures; growth in IBCs eventually culminates in bacterial shedding or the complete exfoliation of the IBC. The underlying dynamics of these phenotypes is hard to capture in the mouse model, but the bladder-chip model allowed us to track the dynamics of bacterial growth within individual IBCs over extended periods of time, providing information on bacterial dynamics within so-called early, middle, and late-stage IBCs. In experiments performed with ampicillin, the heterogeneity in growth rates is evident from the timeseries for three IBCs (*Figure 3G1–G5*, *Figure 3—video 2*). IBCs#1 and 2 subsequently began to shed individual bacteria (*Figure 3G4*), a phenotype not observed in IBC#3.

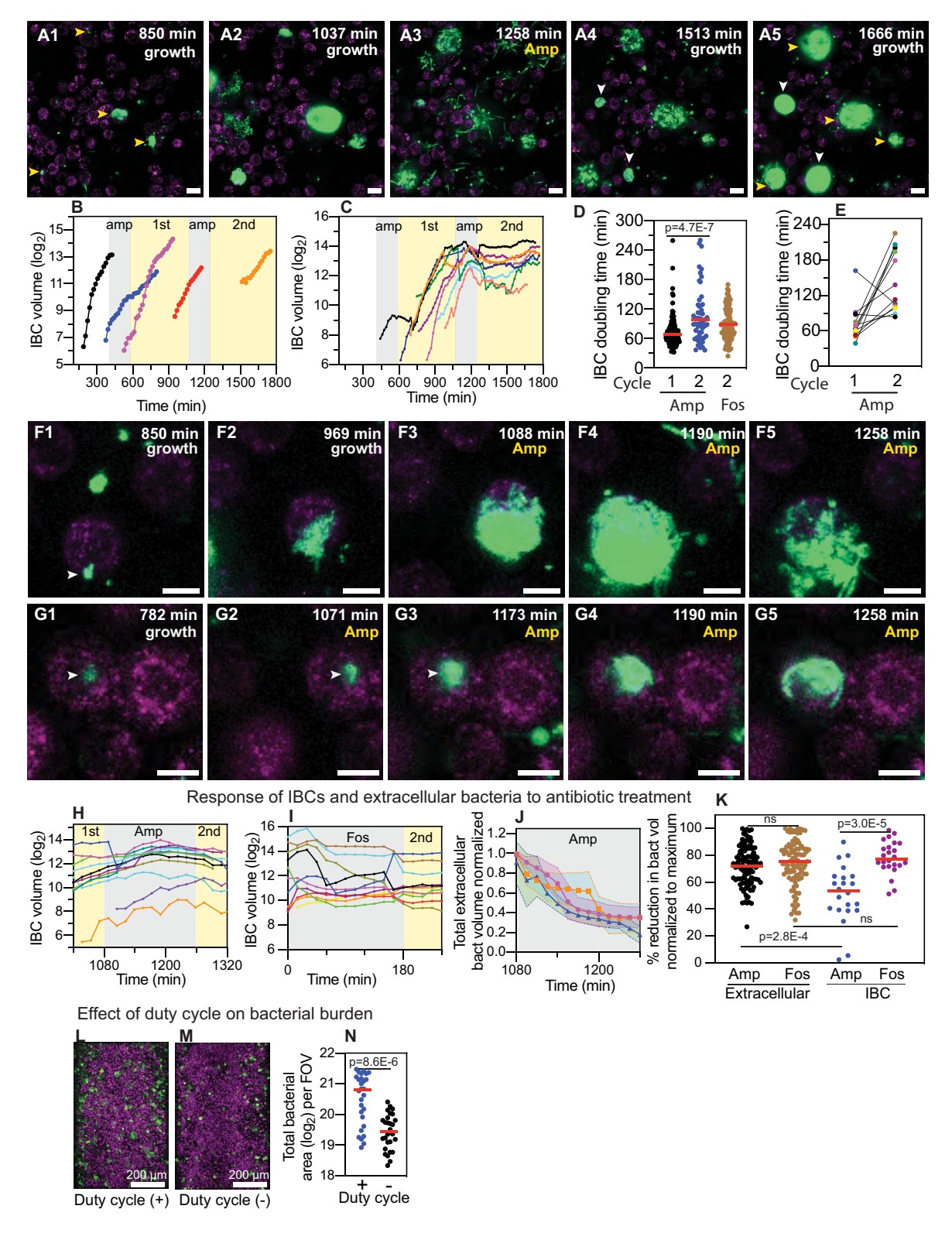

**Figure 4.** IBCs offer a semi-protective niche that delays clearance of bacteria by antibiotics. (**A1–A5**) Bacteria persist and grow in IBCs despite antibiotic treatment. Snapshots show the growth of four IBCs with variable growth rates (**A1–A2**) indicated by yellow arrowheads. Ampicillin treatment eliminates some but not all of the bacteria within each IBC (**A3**). Growth resumes at all sites in (**A4, A5**), formation of new IBCs in the second growth cycle is indicated by white arrowheads. (**B**) Plots of logarithm of bacterial volume within five separate IBCs demonstrates exponential bacterial growth. IBCs are

*Figure 4 continued on next page*

*Figure 4 continued*

seeded dynamically; growth can occur either in the first or second growth cycle, or prior to the administration of antibiotic, or in few cases continues in the presence of the antibiotic. (**C**) Plots of the logarithm of bacterial volume vs. time for n=8 IBCs tracked across two growth phases with an intermediate period of ampicillin treatment. In the growth phases, growth is exponential and bacterial volume continues to increase for up to ca. 120 min after administration of antibiotic before declining due to loss of bacteria. In each case, growth resumes after the antibiotic is removed. (**D**) Scatter plots of the doubling time of bacterial volume in IBCs as measured in the first growth cycle (n=102) and the second growth cycle (n=59) from n = 3 bladder-chips. Growth in the second cycle is significantly slower (p=4.7E-7), red line represents the median value. (**E**) Doubling time of bacterial volume in IBCs in the first and second growth cycle for some of the IBCs in (**D**) that survived the antibiotic treatment. In n=16 out of 18 instances, growth is slower in the second growth cycle. (**F1–F5**) High-resolution time-series that highlights bacterial growth within an IBC prior to (**F1, F2**) and after (**F3–F5**) administration of ampicillin. Some bacteria within this IBC are subsequently eliminated (**F5**). (**G1–G5**) High-resolution time-series that highlights bacterial growth within an IBC prior to (**G1, G2**) and after (**G3–G5**) administration of ampicillin. The bacterial volume within this IBC is not diminished by antibiotic treatment (**G5**). (**H**) Plots of logarithm of bacterial volume within n=11 IBCs before, during, and after the second round of ampicillin treatment. (**I**) Plots of logarithm of bacterial volume within n=9 IBCs before, during, and after the second round of fosfomycin treatment. (**J**) Plot of the volume of extracellular bacteria upon antibiotic administration from n=103 fields of view, each of which was 206 x 206 $\mu m^2$ in size, from n=3 bladder-chips. The bacterial volumes are normalized to the volume immediately prior to the antibiotic administration. (**K**) Scatter plot of the extracellular bacterial volume (n=105) and bacterial volume within IBCs (n=22) from n = 3 bladder-chips after ampicillin treatment and extracellular bacterial volume (n=101) and bacterial volume within IBCs (n=23) from n=1 bladder-chip after fosfomycin treatment as a fraction of the maximum bacterial volume prior to antibiotic treatment. Red line represents the median value, p=2.8E-4 (extracellular amp vs IBC amp), p=3.0E-5 (IBC amp vs. IBC fosfomycin) calculated using Kruskal-Wallis ANOVA Test, 'ns' represents p>0.05 (**L, M**) Representative images from the epithelial face of the infected bladder-chips with or without duty cycle, fixed at ca. 13.5 hr after UPEC infection and 6 hr into the first growth cycle. (**N**) Scatter plots of the logarithm of the total bacterial area across n=14 fields of view from infected bladder-chips with (n=2) and without (n=2) the application of the duty cycle, p = 8.6E-6. p-values calculated using a Mann-Whitney test. Yellow rectangular area in **B**, **C**, **H** and **I** represent the 1st and 2nd growth cycles. Gray rectangular area in **B**, **C**, **H**, **I** and **J** represent the rounds of ampicillin treatment. Scale bars, 10 µm in panels **A1–A5**, **F1–F5**, and **G1–G5**.

The online version of this article includes the following video and figure supplement(s) for figure 4:

**Figure supplement 1.** UPEC regrowth on-chip despite two consecutive periods of fosfomycin treatment.
**Figure supplement 2.** UPEC growth in IBCs during ampicillin treatment.
**Figure supplement 3.** UPEC infection leads to a higher bacterial burden in bladder-chips perturbed with duty cycle.
**Figure 4—video 1.** Bacteria within an IBC can persist and grow within an IBC despite the antibiotic treatment (*Figure 4A1–A5*).
https://elifesciences.org/articles/66481#fig4video1
**Figure 4—video 2.** Ampicillin-mediated bacterial clearance is delayed within an IBC (*Figure 4F1–F5*).
https://elifesciences.org/articles/66481#fig4video2
**Figure 4—video 3.** Ampicillin-mediated bacterial clearance is delayed within an IBC (*Figure 4—figure supplement 1A1-A5*).
https://elifesciences.org/articles/66481#fig4video3
**Figure 4—video 4.** Bacteria can continue growing within an IBC for the entire duration of ampicillin treatment (*Figure 4G1–G5*).
https://elifesciences.org/articles/66481#fig4video4
**Figure 4—video 5.** Bacteria can continue growing within an IBC during the ampicillin treatment (*Figure 4—figure supplement 1B1-B5*).
https://elifesciences.org/articles/66481#fig4video5

---

At a later timepoint (*Figure 3G5*) the fates of all three IBCs were dramatically different – the cell containing IBC#1 had completely exfoliated and was removed from the field of the view by flow in the epithelial channel, IBC#2 continued to shed bacteria, whereas growth remained slow in IBC#3. We monitored the eventual fate of n=100 IBCs during the first and the second growth cycles which confirmed that shedding and exfoliation were not mutually exclusive. While some IBCs only shed bacteria (n=28) and others exfoliated without shedding (n=27), a substantial fraction of IBCs showed shedding prior to exfoliation (n=31) and some IBCs displayed neither phenotype (n=14). Furthermore, in addition to the coccoid and rod-shaped morphologies highlighted in *Figure 3C–F*, time-lapse imaging also enabled us to capture morphotypes that occurred relatively rarely such as the intracellular growth of filamentous forms of UPEC (*Figure 3H1–H5*, *Figure 3—figure supplement 3*, *Figure 3—video 3*, *Figure 3—video 4*). In this time series, a filamentous bacterium can be seen and appears to encircle the inner boundary of the cell (*Figure 3H1–H4*) before filling the volume entirely (*Figure 3H5*). Overall, time-lapse imaging confirms that a significant fraction of bacteria within IBCs can survive the first round of antibiotic treatment and are the source for reseeding of both the extracellular bacterial populations as well as the subsequent growth of IBCs in newly infected cells.

# Dynamic persistence of intracellular bacterial communities within uroepithelial cells

A subsequent round of antibiotic treatment provided an opportunity to study the dynamic persistence of bacteria within IBCs in response to an antibiotic profile that was a closer mimic of the periodic antibiotic exposures in vivo. Furthermore, at this stage of the experiment, we were also able to study the responses of bacteria in intermediate and mature late-stage IBCs to antibiotic treatment, which was not possible during the first round of antibiotic administration early in the course of infection. Owing to the complexity of long-term live-cell imaging, we focused on ampicillin treatment in chips reconstituted with the 5637 cells. We corroborated these findings with live-cell imaging and snapshots taken from chips reconstituted with 5637 cells and primary epithelial cells respectively and treated with fosfomycin. Live-cell imaging in chips treated with ampicillin (*Figure 4A1–A5* and *Figure 4—video 1*) shows that bacterial growth after the first round of antibiotic treatment resulted in the formation of many large IBCs with tightly packed bacteria (*Figure 4A1–A2*). Many of the bacteria within each of the four IBCs were eliminated by the antibiotic, but a substantial proportion of bacteria in each IBC nevertheless survived the antibiotic treatment (*Figure 4A3*). Each of these IBCs subsequently regrows when the antibiotic is removed (*Figure 4A4–A5*, yellow arrowheads), and two additional IBCs are seeded during this time (*Figure 4A4–A5*, white arrowheads). Similar results were obtained with chips reconstituted with primary epithelial cells and treated with fosfomycin (lower resolution whole-chip scans in *Figure 4—figure supplement 1A,B*). Overall, the second round of antibiotic administration led to a sharp decline in instances of subsequent bacterial regrowth (*Figure 3—figure supplement 1C*). In all instances, regrowth either occurred directly within IBCs, or was caused by the shedding of bacteria from IBCs to repopulate the extracellular niche and seed new IBCs, highlighting the dynamic stability of this niche and its importance in establishing persistent infection. We therefore performed a careful analysis to quantify bacterial growth rate within IBCs. Growth within IBCs across different time periods was exponential in both epithelial cells (*Figure 4B*) and primary epithelial cells (*Figure 4—figure supplement 1C*). We were able to track a subset of IBCs across two growth cycles (examples in *Figure 4C*). In a majority of instances these revealed exponential growth in the absence of antibiotic, delayed response to antibiotic, and a lag phase after the antibiotic was removed. In general, bacterial growth in IBCs in the second growth cycle was slower than in the first (*Figure 4D*). However, this population level statistic may be influenced by the fact that many of the IBCs monitored in the second growth cycle had been exposed to two rounds of antibiotic treatment. For this sub-population, in n=16 out of n=18 IBCs, growth was slower after the second round of antibiotic administration (*Figure 4E*).

Next, we examined the dynamics of bacterial growth within IBCs during the period of antibiotic treatment in greater detail. In the IBC shown in the timeseries in (*Figure 4F1–F5*, *Figure 4—figure supplement 2A1-A5*, *Figure 4—video 2*, *Figure 4—video 3*), growth before antibiotic administration (*Figure 4F1–F2*) continued for a considerable period after the antibiotic was administered (*Figure 4F3–F4*, *Figure 4—figure supplement 2A1-A3*). Eventually, a reduction in bacterial volume towards the end of the antibiotic administration period was observed (*Figure 4F5*, *Figure 4—figure supplement 2A4-A5*). In contrast, the relatively smaller IBC in the timeseries in *Figure 4G1–G5*, *Figure 4—figure supplement 2B2-B5*, *Figure 4—video 4*, *Figure 4—video 5* continued to grow throughout the period of antibiotic administration. These different dynamics are also captured in the plot of bacterial volume for multiple IBCs before, during, and after the antibiotic treatment (*Figure 4H*). In all cases, bacterial volume continues to increase during a significant period of antibiotic administration and in 2 out of 18 cases, there was no decrease in bacterial volume throughout this period (*Figure 4H*). In contrast, extracellular bacteria likely adherent on the epithelial cells but not internalized within them were rapidly eliminated by combination of antibiotic treatment and flow (*Figure 4J*), in all cases the bacterial volume reduced immediately upon antibiotic administration. Fosfomycin treatment elicited a rapid decline in both extracellular bacterial volume and bacterial volume in IBCs, consistent with the greater cellular penetrance of this antibiotic (*Figure 4I*). This differential clearance of bacteria was therefore niche- and antibiotic-specific, with a significantly higher reduction in the extracellular bacterial volume vs. IBC volume, and a significantly higher reduction in IBC volume with treatment with fosfomycin vs. ampicillin (*Figure 4K*). However, elimination of bacteria within IBCs was highly heterogenous, and regrowth localized to IBCs was observed even in fosfomycin treated chips (*Figure 4—figure supplement 1A5, B5*). Another consequence of this

differential clearance was that after the second round of antibiotic treatment, regrowth was observed in only a very small fraction of non-IBC UPEC (*Figure 3—figure supplement 1C*). Protection of bacteria within IBCs therefore has a direct outcome of enabling the reseeding of infection at other locations within the epithelial monolayer.

Lastly, we sought to determine if the bladder duty cycle altered the dynamics of UPEC infection. A comparison between infected bladder-chips with and without the duty cycle (*Figure 4L–N*, *Figure 4—figure supplement 3*) revealed a significant increase in the bacterial burden when the duty cycle was implemented (*Figure 4N*). These results suggest that the bladder filling and voiding cycle influence the uptake and proliferation of UPEC, possibly through physiological changes in the epithelial cells in response to applied strains (*Apodaca, 2004*; *Carattino et al., 2013*; *Truschel et al., 2002*; *Wang et al., 2005*). In turn, they suggest a deeper connection between the physiology of mechanically active organs such as the bladder and the processes of infection, which can be characterized in detail using this bladder-chip model.

## Discussion

In our human bladder-chip model, we recapitulate key aspects of bladder physiology relevant to the study of the earliest stages of UPEC infection and IBC formation. Therefore, superficial bladder epithelial cells, the first cell-type usually infected by UPEC, were cultured both under flow in diluted pooled urine and in co-culture with bladder microvascular endothelial cells. The ability to apply a cyclic mechanical stretch to the PDMS membrane, originally designed to mimic the breathing motion in the lung (*Huh et al., 2010*) or the peristaltic motion in the gut (*Kim and Ingber, 2013*), has been adapted here to mimic the slow expansion and rapid contraction of the micturition cycle. We demonstrate the ability to perform multiple duty cycles while simultaneously imaging the infected device via long-term time-lapse imaging, a technical advance that is difficult to achieve with bladder explants (*Justice et al., 2004*) or other in vitro studies of UPEC infection of bladder epithelial cells (*Andersen et al., 2012*; *Horsley et al., 2018*; *Iosifidis and Duggin, 2020*; *Smith et al., 2006*). Using this approach, we found that the total bacterial burden inside infected bladder-chips was significantly higher at a late stage of infection if a duty cycle was applied, unlike the non-significant influence of cyclic stretching on Shigella infection of intestinal-chips under flow conditions (*Grassart et al., 2019*). The exact mechanisms underlying this phenomenon remain to be explored, but there is an increasing understanding of the role of mechanical forces in regulating innate immune function (*Solis et al., 2019*). Our results showcase the ability of the bladder-chip model to capture these interactions between mechanical function, physiology, and infection, unlike other infection models reported thus far.

A primary focus of in vitro models developed so far has been to probe specific aspects of UPEC infection, such as the role of the stratified bladder architecture (*Horsley et al., 2018*) or the effects of micturition on IBC formation (*Andersen et al., 2012*; *Iosifidis and Duggin, 2020*). However, the migration of immune cells into the bladder is difficult to reproduce in these models, and in many of these systems, live-cell imaging remains technically challenging (*Horsley et al., 2018*; *Smith et al., 2006*). Furthermore, the models do now allow mechanical manipulations of the cellular co-culture. In that sense, the bladder-chip model complements these existing approaches by providing these additional functionalities. Further development, potentially through the combination of organoid and organ-on-chip approaches, could lead to the development of a fully stratified uroepithelium on-chip.

Neutrophils are the first responders to UPEC infection (*Abraham and Miao, 2015*; *Haraoka et al., 1999*), and neutrophil migration involves a series of steps that commences with attachment to the endothelium under flow, migration on the endothelial surface, diapedesis across the epithelial-endothelial cell barrier and movement toward the site of infection (*Ley et al., 2007*; *Nourshargh et al., 2010*). Human neutrophils in the bladder-chip model recapitulate all these phenotypes and perform rapid diapedesis to sites of infection. On the epithelial side, neutrophils formed swarm-like aggregates, some of which generated myeloperoxidase and neutrophil elastase positive NET-like structures that extend across many tens of microns (*Metzler et al., 2011*; *Papayannopoulos et al., 2010*), indicative of potent anti-microbial activity. However, neutrophil control of infection on-chip is partial; this is possibly due to the unrestricted growth of large numbers of extracellular bacteria and in some instances, exacerbated by the loss of some neutrophils due to

flow in the epithelial channel. Another contributing factor could potentially be the architecture of the PDMS membrane, which permits neutrophil diapedesis only at fixed spatial locations on-chip. Neutrophil migration may also be impacted by the relative stiffness of the PDMS membrane. Nevertheless, the demonstration of NET formation is consistent with the occurrence of these structures both in the mouse model (*Ermert et al., 2009*) as well as in the urine of infected patients (*Yu et al., 2017*) and suggests that the model is able to recapitulate important aspects of disease. It is also an important advance for the use of organ-on-chip approaches to recapitulate NET formation in infectious diseases.

IBCs form immediately after infection and prior to upon neutrophil administration. However, identification of IBCs architectures with a high degree of confidence was only possible after an initial treatment with a high dose of antibiotic that eliminated extracellular planktonic bacteria and improved the optical imaging of bacteria attached to or within epithelial cells. This methodology enabled us to capture the full cycle of IBC growth from few bacteria to a large biofilm and subsequent release via shedding and filamentation. Importantly, the compact nature of the device allowed us to image multiple IBCs concurrently on the same chip with a high temporal resolution while simultaneously mimicking the bladder filling and voiding cycles, which is difficult to achieve in bladder explants (*Justice et al., 2004*) or bladder monoculture in vitro model systems (*Andersen et al., 2012*; *Iosifidis and Duggin, 2020*). Our observations reiterate the highly dynamic nature of these structures; growth was asynchronous and heterogenous and outcomes included bacterial shedding, exfoliation and filamentation (*Hunstad and Justice, 2010*; *Justice et al., 2004*; *Scott et al., 2015*). Notably, shedding and exfoliation were not mutually exclusive for the time-period of our observations – we report examples of IBCs that shed bacteria and contract in volume before exfoliation.

While IBCs are clearly acknowledged as critical players in early infection, the contribution towards promoting persistence is not completely understood. For example, *Blango and Mulvey, 2010* showed that incubation of bladder explants containing IBCs with a high dose of antibiotics for a period of 18 hr resulted in a substantial sterilization of these structures and concluded that other populations, notably quiescent reservoirs (*Mysorekar and Hultgren, 2006*) might play a greater role in the establishment of persistent populations. However, the pharmacokinetic profiles of most antibiotics in the host are not time invariant. For example, in the case of ampicillin, a standard regimen of ampicillin treatment is typically a bolus of 250–500 mg of antibiotic administered every 6 hr. Within the serum, ampicillin concentration peaks at a $C_{max} \sim$ 3–40 µg/ml, which is between 1.5 and 20-fold the MIC as measured in human serum (*Bryskier, 2020*; *Putrinš et al., 2015*). We therefore chose concentrations that captured these antibiotic exposures (we used ampicillin at 40x MIC measured in the endothelial cell medium). Ampicillin concentrations in the blood rapidly decay with a half-life of between 60 and 90 min with a characteristic pharmacokinetic/pharmacodynamic profile. This period is modelled well by our experimental protocol where phases with high concentration of ampicillin are interspersed with periods with no antibiotic. We are therefore able to model the delivery of two consecutive doses of antibiotic and find that IBCs offer substantial protection against sterilization by a short duration of a high-dose antibiotic treatment and in many instances, bacterial regrowth after two successive rounds of antibiotic administration. Notably, in smaller 'early IBCs', bacterial growth continues throughout the period of ampicillin administration, suggesting that the intact nature of the cell membrane likely diminishes the effect of the drug. Although fosfomycin has a different pharmacokinetic/pharmacodynamic profile it has enhanced cell penetrability. Nevertheless, our results show bacteria in IBCs can regrow following two consecutive 3 hr exposures fosfomycin at concentrations reported in urine (*Bergan, 1990*). Together, these results suggest that in UTI infection in patients, IBCs may continue to play a role in reseeding sites of infection for a considerable period after the commencement of antibiotic treatment. This has particularly important implications with regard to the compliance of antibiotic use as proliferating IBCs could rapidly re-seed sites of infection throughout the bladder if antibiotic doses are missed. These unique capabilities of the bladder-chip to realistically model antibiotic treatment regimens for IBCs can be leveraged in the future to screen compounds (*Spaulding et al., 2017*) and identify optimal antibiotic treatments regimens that can eliminate persistent bacterial populations in IBCs or alter the host-pathogen interaction dynamic in UTIs.

In summary, the bladder-chip model incorporates aspects of bladder physiology highly relevant to early UPEC infection in a platform amenable to long-term live-cell imaging as well as for the administration of antibiotics and therapeutics in a physiologically relevant manner. Our results

establish the suitability of this model for immunological and drug-delivery studies and show that IBCs are highly dynamic structures that offer substantial protection from antibiotic clearance for an extended period of time.

# Materials and methods

## Key resources table

| Reagent type (species) or resource | Designation | Source or reference | Identifiers | Additional information |
|---|---|---|---|---|
| Strain, strain background (Uropathogenic *Escherichia coli*) | CFT073 | PMID:2182540 | NCBI: txid199310 | originally isolated from a pyelonephritis patient and provided by Prof. H.L.T. Mobley, University of Michigan, USA |
| Recombinant DNA reagent | pZA32-YFP (plasmid) | PMID:9092630 **Lutz and Bujard, 1997** | | |
| Strain, strain background (Uropathgenic *Escherichia coli*) | CFT073-pZA32-YFP, CFT073 | this paper | | this study |
| Cell line (*Homo sapiens*) | HTB-9 bladder epithelial cells | ATCC | Cat#:5637 | |
| Cell line (*Homo sapiens*) | HMVEC-Bd – Human Bladder Microvascular Endothelial Cells | Lonza | Cat#:7016 | |
| Biological sample (*Homo sapiens*) | Human primary bladder epithelial cells | Cell Biologics | Cat#:H-6215 | |
| Other | RPMI-1640 medium | ATCC | Cat#:30–2001 | |
| Other | Gibco RPMI 1640 Medium, no phenol red | Thermofisher | Cat#:11835063 | |
| Other | GibcoFetal Bovine Serum, Premium Plus | Thermofisher | Cat#: A4766801 | |
| Other | Complete Human Epithelial Cell Medium with growth factor supplement | Cell Biologics | Cat#: H6621 | |
| Other | EGM-2 MV Microvascular Endothelial Cell Growth Medium-2 BulletKit | Lonza | Cat#:CC-3202 | |
| Other | EBM-PRF Endothelial Medium Phenol-red free, 500 ml | Lonza | Cat#:CC-3129 | |
| Other | EGM-2 Endothelial SingleQuots Kit | Lonza | Cat#:CC-4176 | |
| Other | Gibco Antibiotic-Antimycotic (100X) | Thermofisher | Cat#:15240062 | |
| Other | Gibco Trypsin-EDTA (0.05%), phenol red | Thermofisher | Cat#:25300054 | |
| Drug | Chloramphenicol | Sigma-Aldrich | Cat#:C1919-25G | 34 mg/ml in ethanol (stored at −20˚C) |
| Drug | Ampicillin | Sigma-Aldrich | Cat#: A9518-5G | 50 mg/ml in ddH$_2$O (stored at −80˚C) |
| Other | Pooled human female urine | Golden West Diagnostics | Cat#: OH2010-pH | |
| Other | Phosphate Buffered Saline | Thermofisher | Cat#: 10010056 | |
| Other | Gibco HEPES | Thermofisher | Cat#: F2006 | |
| Other | Invitrogen DAPI | Thermofisher | Cat#: D1306 | 5 mg/ml in DMSO |

*Continued on next page*

*Continued*

| Reagent type (species) or resource | Designation | Source or reference | Identifiers | Additional information |
|---|---|---|---|---|
| Other | LB (Luria broth base, Miller's modified) | Sigma-Aldrich | Cat#: L1900-1KG | |
| Other | Organ-chips (Standard Research Kit - 24 per Pack) | Emulate | RE1000001024 | |
| Other | CellMask Orange Plasma membrane Stain | Thermofisher | Cat#: C10045 | 1 µM in cell media |
| Other | HCS CellMask Deep Red Stain | Thermofisher | Cat#: H32721 | 2 µM in PBS |
| Other | CellTracker Deep Red Dye | Thermofisher | Cat#: C34565 | 1 µM in cell media |
| Other | Elveflow OB1 Pressure Controller OB1 Base MkIII+ | Elveflow | Cat#: OB1MKIII+-MIX- | connected to 6 bar compressed air channel |
| Other | Elveflow OB1 Pressure Controller OB1 MkIII Channel-900/1000 | Elveflow | Cat#: OB1-Dual- | connected to Elveflow OB1 Pressure Controller OB1 Base MkIII+ |
| Other | Diaphragm moist gas vacuum pump | KNF Neuberger | Cat#:LABOPORT UN 820.3 FT.40P | connected to Elveflow OB1 Pressure Controller OB1 MkIII Channel-900/1000 |
| Antibody | Anti-EpCAM (rabbit polyclonal) | Abcam | Cat#: ab71916 | IF (1:100) in 1% BSA |
| Antibody | Anti-PECAM1 or Anti-CD31 (mouse monoclonal) | Abcam | Cat#: ab24590 | IF (1:100) in 1% BSA |
| Antibody | Anti-CK7 (rabbit monoclonal) | Abcam | Cat#: ab209599 | IF (1:100) in 1% BSA |
| Antibody | Anti-CK8 (rabbit monoclonal) | Abcam | Cat#: ab192468 | IF (1:100) in 1% BSA |
| Other | Alexa Fluor 555 Phalloidin | Thermofisher | Cat#: A34055 | 1 µM in PBS |
| Antibody | Anti-VE-Cadherin (rabbit polyclonal) | abcam | Cat#: ab33168 | IF (1:100) in 1% BSA |
| Antibody | Anti-Uroplakin3a (mouse monoclonal) | Santa Cruz | Cat#: sc-166808 | IF (1:100) in 1% BSA |
| Antibody | Anti-CK1 (mouse monoclonal) | Thermofisher | Cat#: MA1-06312 | IF (1:100) in 1% BSA |
| Antibody | Anti-E-Cadherin (mouse monoclonal) | Thermofisher | Cat#: MA5-14408 | IF (1:100) in 1% BSA |
| Antibody | Anti-ZO-1 (rabbit polyclonal) | abcam | Cat#: ab216880 | IF (1:100) in 1% BSA |
| Antibody | Anti-myeloperoxidase (rabbit polyclonal) | abcam | Cat#: ab9535 | IF (1:100) in 1% BSA |
| Antibody | Anti-neutrophil elastase (rabbit polyclonal) | abcam | Cat#: ab68672 | IF (1:100) in 1% BSA |
| Antibody | Anti-CD15 (mouse monoclonal) | abcam | Cat#: ab665 | IF (1:100) in 1% BSA |
| Antibody | Donkey anti-Mouse IgG (H+L) Highly Cross-Adsorbed Secondary Antibody, Alexa Fluor 647 | Thermofisher | Cat#: A-31571 | IF (2 µg/ml) |
| Antibody | Donkey anti-Rabbit IgG (H+L) Highly Cross-Adsorbed Secondary Antibody, Alexa Fluor 647 | Thermofisher | Cat#: A-31573 | IF (2 µg/ml) |
| Antibody | Goat anti-Mouse IgG (H+L) Highly Cross-Adsorbed Secondary Antibody, Alexa Fluor 488 | Thermofisher | Cat#: A-11029 | IF (2 µg/ml) |

*Continued on next page*

*Continued*

| Reagent type (species) or resource | Designation | Source or reference | Identifiers | Additional information |
|---|---|---|---|---|
| Antibody | Donkey anti-Rabbit IgG (H+L) Highly Cross-Adsorbed Secondary Antibody, Alexa Fluor 568 | Thermofisher | Cat#: A10042 | IF (2 µg/ml) |
| Antibody | Donkey anti-Mouse IgG (H+L) Highly Cross-Adsorbed Secondary Antibody, Alexa Fluor 568 | Thermofisher | Cat#: A10037 | IF (2 µg/ml) |
| Peptide, recombinant protein | Fibronectin from human plasma | Sigma-Aldrich | Cat#: F1056 | 500 µg/ml in ddH$_2$O |
| Peptide, recombinant protein | Native Collagen, Bovine dermis | AteloCell | Cat#: IAC-50 | 5 mg/ml |
| Peptide, recombinant protein | Human IL-1α, research grade | Miltenyi Biotec | Cat#:130-093-894 | 1 µg/ml |
| Peptide, recombinant protein | Human IL-1β, research grade | Miltenyi Biotec | Cat#:130-093-895 | 1 µg/ml |
| Peptide, recombinant protein | Human IL-6, research grade | Miltenyi Biotec | Cat#:130-093-929 | 1 µg/ml |
| Peptide, recombinant protein | Human IL-8, research grade | Miltenyi Biotec | Cat#:130-122-354 | 1 µg/ml |
| Commercial assay or kit | MACSxpress Whole Blood Neutrophil Isolation Kit, human | Miltenyi Biotec | Cat#:130-104-434 | |
| Software, algorithm | Imaris 9.5.1 | Bitplane | | |
| Other | Gas chamber for stages with k-frame insert (160x110mm) - magnetic model with sliding lid. | okolab | Cat#: H201-K-FRAME | |
| Other | Custom holder for 24 mm x 60 mm coverslip | okolab | Cat#:1x24by60-M | |
| Other | µ-Slide 8 Well | ibidi | Cat#:80826 | |
| Other | Masterflex PharMed tubing, 0.89 mm ID, 100 ft | Cole-Palmer | Cat#:GZ-95709–26 | |
| Other | Masterflex Transfer Tubing, Tygon 0.76 mm ID | Cole-Palmer | Cat#:GZ-06419–03 | |
| Other | 1.30 x 0.75 x 10 mm metallic tubes | Unimed | Cat#:9084 / 200.010-A | |
| Other | 1.00/0.75 x 20 mm metallic tubes | Unimed | Cat#:200.010-A | |
| Other | Aladdin programmable pump | WPI | Cat#: PUMP-NE-1000 | |

## Cell lines

5637 human bladder epithelial carcinoma cell line (procured from ATCC, HTB-9TM). Cell line tested negative for mycoplasma contamination.

## Cell culture of human bladder epithelial and bladder endothelial cells

The 5637 human bladder epithelial carcinoma cell line (ATCC, HTB-9) was cultured in RPMI 1640 medium supplemented with 10% Fetal Bovine Serum (FBS) as recommended by the supplier. Human Primary Bladder Epithelial cells (Cell Biologics, H-6215) were cultured in complete epithelial cell

medium supplemented with growth factors as recommended by the supplier. Human Bladder Microvascular Endothelial cells (HMVEC-Bd) (Lonza, CC-7016) were cultured in Lonza EGM-2 MV BulletKit medium. Both the epithelial and endothelial cells were cultured in their respective medium supplemented with 1X Antibiotic-Antimycotic solution. Epithelial cells were passaged by detachment with 0.05% Trypsin at 37°C for 3–5 min followed by neutralization of trypsin with RPMI 1640 medium/10% FBS. Bladder endothelial cells were passaged and split as recommended by the supplier. The cells used in all the experiments were at ten passages or fewer. The cell lines were tested routinely for mycoplasma contamination during passaging.

## UPEC culture for infection of bladder epithelial cells in the bladder-chip

Uropathogenic *Escherichia coli* (UPEC) strain CFT073 was originally isolated from a pyelonephritis patient (*Mobley et al., 1990*) and provided by Prof. H.L.T. Mobley, University of Michigan, USA. A derivative strain expressing yellow fluorescent protein (YFP) was generated by electroporation of CFT073 with the episomal plasmid pZA32-YFP (*Lutz and Bujard, 1997*), as described earlier (*Dhar et al., 2015*). To induce expression of type 1 pili, UPEC was grown in LB media containing 25 µg/ml chloramphenicol under non-shaking conditions at 37°C for 2 days prior to the experiment, to achieve a stationary phase culture ($OD_{600}$=1.5 and corresponding to a concentration of 1.5 x $10^9$ bacteria/ml). The bacteria were diluted 10-fold to a final concentration of (1.5 x $10^8$ cells/ml) and resuspended in a solution of pooled human female urine (Golden West Diagnostics Catalogue OH2010-pH) diluted 10-fold in Phosphate Buffered Saline.

## Recapitulation of human bladder physiology in a human bladder–chip device

Bladder-chip devices made of polydimethylsiloxane (PDMS) were purchased from Emulate (Boston, USA). The dimensions of the microfluidic device were as follows: width of the channels- 1000 µm, height of the upper channel- (1000 µm, lower channel – 250 µm). For extracellular matrix (ECM) coating, a coating solution consisting of the ER-1 compound (Emulate) dissolved in ER-2 solution at 0.5 mg/ml (Emulate) was introduced in both apical and vascular channels and the chips were subsequently activated by exposing the bladder-chip for 20 min under UV light. The channels were then rinsed with fresh coating solution and the protocol was repeated once again. The channels of the bladder-chip were then washed thoroughly with PBS before incubation with an ECM solution of 150 µg/ml bovine collagen type I and 30 µg/ml fibronectin from human plasma in PBS buffered with 15 mM HEPES solution for 1–2 hr at 37°C as described previously (*Thacker et al., 2020*). If not used directly, coated chips were stored at 4°C and pre-activated before use by incubation for 30 min with the same ECM solution at 37°C. Three days before the day of the experiment, HMVEC-Bd cells were seeded into lower channel of the inverted bladder-chip device at 0.5 million cells/ml. Two days before the experiment, 5637 cells were seeded into upper channel at 5 million cells/ml. Prior to the infection experiment, bladder endothelial and bladder epithelial cells were cultured in their respective medium supplemented with the 1X Antibiotic-Antimycotic solution. The antibiotics were removed prior to the experiment.

## Characterization and immunostaining of human bladder epithelial and human bladder epithelial cells in ibidi wells and human bladder-chip

Human bladder epithelial and bladder endothelial cells were cultured inside ibidi µ−Slide eight wells for one day. The cells were subsequently fixed with 4% paraformaldehyde (PFA) for 30 min at room temperature. Fixed cells were then washed three times with 200 µl of PBS to remove residues of PFA, permeabilized with 0.15% Triton X-100 for 15 min; washed three times with 200 µl of PBS to remove residues of detergent and then incubated in a blocking solution of 1% BSA in PBS for 1 hr. The cells were then incubated with primary antibodies (anti-EpCAM, anti-PECAM-1, anti-CK7, anti-CK8, anti-VE-Cadherin, anti-Uroplakin3a, anti-CK1, anti-E-Cadherin, anti-ZO-1, anti-myeloperoxidase, anti-neutrophil elastase, anti-CD15) at a dilution of 1:100 in an antibody incubation buffer comprising 1% BSA and 0.01% Triton-100 in PBS. The ibidi eight-wells were subsequently washed three times with PBS for 10 min each. Incubation with secondary antibody (Donkey anti-Mouse Alexa Fluor 647, Donkey anti-Mouse Alexa Fluor 568, Goat anti-Mouse Alexa Fluor 488, Donkey anti-Rabbit Alexa Fluor 647, Donkey anti-Rabbit Alexa Fluor 568, Donkey anti-Rabbit Alexa Fluor 488) at a

concentration of 2 µg/ml in antibody incubation buffer was subsequently performed for 1 hr at room temperature. Excess antibody was removed by washing three times with PBS for 10 min each. Cell nuclei were stained with DAPI (5 µg/ml) solution in PBS for 30 min. Ibidi eight-wells were washed three times to remove unbound DAPI. Cells were covered with appropriate volume of PBS until imaging. Cells were imaged with 63X oil objective on Leica SP8 confocal microscope. Images were deconvolved using SVI Huygens (Quality, 0.05; Iterations, 40).

For characterization of the cell types on-chip, bladder-chips with co-culture of bladder epithelial and bladder endothelial cells were kept under pooled diluted urine and EBM2 medium using P200 pipette tips for 2 days prior to PFA fixation. The cells in the bladder-chip were then immunostained following the protocol described above. Images were acquired with Leica HC FLUOTAR 25X (NA 0.95) multi-immersion objective on Leica SP8 confocal microscope.

## Characterization of strain-pressure curve in the bladder-chip model

Elveflow OB1 MK3 – Microfluidic flow control system was used to control the negative pressure applied to the human bladder-chip. The control system was connected to the compressed air line (6 bar) for the positive pressure and diaphragm vacuum pump for the negative pressure. A negative pressure (0 to −900 mbar) with a step function of −100mbar was subsequently applied to the vacuum channels in the bladder-chip using a Pressure Controller (Elveflow OB1 pressure controller). For these experiments, both bladder epithelial and endothelial cells were seeded on the respective sides in the devices. This experiment was performed with the chip maintained on the stage of the microscope and a brightfield image was acquired at each step increase of −100mbar, on Leica SP8 confocal microscope. The PDMS inter-pore-to-pore distance was measured for 14 pore-to-pore combinations at each input of negative pressure. Linear fitting was performed using GraphPad Prism (version 9).

## Mimicking bladder filling and voiding cycle in human bladder-chip

The stretching of the human bladder-chip was done using the Elveflow OB1 MK3 – Microfluidic flow control system connected to the compressed air line (6 bar) for the positive pressure and diaphragm vacuum pump for the negative pressure. Human bladder-chip in the relaxed (*voided bladder*, 0% strain) state was slowly perturbed with a linear ramp function to reach a stretched (*filled bladder*, 10% strain) state over a period of 2 hr. Linear strain of 10% strain was achieved by application of ca. 520–530 mbar negative pressure in the vacuum channels. This period corresponded to the *filling bladder* state in *Figure 1D*. Bladder-chip was kept under stretched (*filled bladder*, 10% strain) state over the subsequent 2 hr. Micturition or urination was recapitulated by rapidly reducing the applied strain on the bladder-chip from the stretched (*filled bladder, 10% strain*) state to relaxed (*voided bladder*, 0% strain) state over a period of 2 min. Subsequently, the bladder-chip was maintained with no negative pressure applied in the vacuum channel under relaxed (*voided bladder*, 0% strain) state from (4:02 hr to 6 hr). This 6 hr bladder filling and voiding cycle was repeated continuously for the rest of the experiment.

## Experimental setup and imaging parameters for time lapse imaging for UPEC infection in the human bladder-chip devices

The medium perfusion inside the bladder-chip device was achieved with the Aladdin syringe pumps. The syringes with respective media were connected to the bladder-chip via gas impermeable PharMed tubing (inner diameter = 0.89 mm, Cole palmer GZ-95809–26) along with longer transparent Tygon tubings (internal diameter of 0.76 mm, Masterflex transfer tubing, Cole palmer). The Pharmed tubing was connected to the inlet and outlet of the bladder-chip with 1.30 x 0.75 x 10 mm metallic tubes (Unimed) and transparent tubing with 1.00/0.75 x 20 mm metallic tubes (Unimed). The bladder-chip connected to external sources of flow was then mounted onto a 24 x 60 mm No. 1 glass coverslip for microscopy imaging. When required for stretching experiments, tubing was also connected to the stretching channels on either side of the main channel. The connected device was subsequently assembled inside a temperature-controlled microscope environmental chamber at 37° C supplemented with 5% $CO_2$ (OKO labs). Time-lapse imaging was conducted using a Leica HC FLUOTAR 25X (NA 0.95) multi-immersion objective within custom made environmental chamber set at 37°C for infusion with syringe pumps. Water was pumped to the ring around the water objective

at 9 Hz with pumping duration of 9 s and pumping interval 30 min, controlled by SRS software (HRZ=9, VPP=95). The autofocus mode (best focus, steps = 9, range = 30 µm) was used to maintain the optical focus on the apical side of the PDMS membrane. The experiments were monitored frequently to ensure that the optical focus was maintained, and the experiment was halted and restarted if the focus was lost.

To enable rapid 3-D imaging across multiple spatial locations on-chip, we utilized the capability of the white light laser on the Leica SP8 confocal microscope to image at multiple wavelengths simultaneously. The excitation wavelengths were grouped into two sequences to minimize the spectral overlap. In the first sequence, laser emission at 555 nm was used to excite the CellMask Orange stain in the bladder epithelial cells. In the second laser excitation sequence, laser emission at 500 nm and 630 nm were used to excite the YFP within the bacteria and the cytoplasmic CellTracker Deep Red in the human neutrophils. Images were acquired with a scan speed of 400–700 Hz and a zoom factor of 2.25 (206.67 µm x 206.67 µm) resulting in an XY resolution of 450 nm depending on the number of pixels acquired per field of view. Z-stacks were acquired with 1 µm step sizes. Time lapse images were acquired with interval duration of ca. 15–17 min, in a subset of experiments this was further reduced to 7.5 min by imaging with the second sequence only at a z-step size of 2 µm.

## Time lapse imaging for UPEC infections in the human bladder-chip devices

### Pre-infection stage

The time-course of the entire experimental protocol is shown in a schematic in *Figure 1E*. The bladder-chip device was perfused with diluted urine in the apical channel and EBM Endothelium phenol-red free medium supplemented with EGM-2 endothelial SingleQuots kit in the vascular channel. Prior to the commencement of infection, the chip was maintained at homeostasis and the epithelial cells were imaged for a period of two hours.

### Infection stage

Stationary phase UPEC in diluted pooled urine at a concentration of 150 million cells/ml were flowed through the apical channel of the device at 1200 µl/hr for 1.5 hr. During this period, EBM2 media was flowed through the endothelial channel at 600 µl/hr.

### Bacterial washout and neutrophil introduction stage

Next, the syringe connected to the apical channel was replaced with a fresh syringe containing pooled diluted urine, this solution was then flowed in the apical channel over the epithelial layer of the chip at a flow rate of 600 µl/hr over the next 3 hr. This allowed for the continuous removal of extracellular planktonic bacteria in the apical channel. At the same time, human neutrophils were introduced into the endothelial channel of the bladder-chip via flow. A solution of human neutrophils at a density of 2 million cells/ml isolated via negative selection was flowed through the vascular channel for 3 hr in EBM2 medium with at higher shear stress of η=1.0 dyne/cm$^2$ to enhance neutrophil attachment to endothelial cells.

### Neutrophil diapedesis stage

Diapedesis of human neutrophils to epithelial side and subsequent interactions of neutrophils with UPEC was observed for the subsequent 3 hr since introduction of neutrophils into the vasculature side of the chip. During this period, the flow rate in the apical and vascular channels were maintained at 600 µl/hr and 3000 µl/hr, respectively. During this period, in a subset of experiments, only the channels that were part of the second laser scanning sequence were imaged. Images of the CellTracker Orange dye for epithelial cell identification were not acquired (first laser scanning sequence). This enabled a number of fields of view to be captured with an enhanced temporal resolution and a frame rate of up to 7.5 min. For experiments studying the formation of NETs, the experiment was halted at this stage and the infected chips were fixed, permeabilized, blocked, and immunostained with anti-myeloperoxidase (abcam) or anti-neutrophil elastase (abcam) antibodies using the procedure described earlier. For all other experiments, live imaging continued to the subsequent antibiotic treatment phase.

## Antibiotic treatment and growth cycles

Thereafter, syringes connected to both apical and vascular channels were changed and ampicillin at 250 µg/ml (used at ~40-fold over the minimum inhibitory concentration (MIC) of ampicillin against UPEC grown in EBM2 medium) was introduced both in the diluted urine and the EBM2 media perfused into the apical and vascular channels, respectively, for 3 hr at a flow rate of 600 µl/hour. This was the first ampicillin treatment cycle; whose purpose was to eliminate extracellular bacteria in the apical channel and allow intracellular bacterial colonies (IBCs) to be identified. The urine and EBM2 media were subsequently switched in the respective channels with antibiotic-free medium for next 8 hr to allow for IBC growth within epithelial cells. During this period, the flow within the apical channel was maintained, to remove bacteria that grew extracellularly either because they survived the antibiotic treatment or that were released from infected epithelial cells. The ampicillin and growth cycle were then repeated, to allow the assessment of the response of bacteria within IBCs to antibiotic treatment as well as characterize the subsequent regrowth.

## Image analysis of confocal live-cell images

Image analysis was performed with Bitplane Imaris 9.5.1. The time-lapse imaging stack included five channels: uroepithelial cells (epithelial channel), transmitted light (bright field channel), UPEC (bacterial channel), neutrophils (neutrophil channel), and transmitted light (bright-field channel). Neutrophils were identified via the spot detection algorithm in Imaris used on the images from the neutrophil channel with the following parameters (Size, 8 µm; Quality, 4 to 8).

Swarms of neutrophils generate dense aggregates that are ill-suited to quantification with the spot detection algorithm. To quantify the size of the swarms, we therefore segmented images in the neutrophil channel to generate surfaces via the automatic segmentation tool in Imaris with the following parameters (Threshold, 10; Smooth Surfaces Detail, 0.5 or 1.0 µm). Unfortunately, the time resolution was insufficient to track individual neutrophils over time particularly since neutrophils formed small clumps and rapidly changed their cell shapes.

For UPEC volume inside IBCs, necessary 3D volume was cropped in Imaris to ease image analysis. Total bacterial volume inside IBC was detected by creating the surface (Threshold, 15; Smooth Surfaces Detail, 0.5 or 1 µm) on UPEC channel.

In cases of measuring extracellular UPEC growth, surface was generated (Threshold, 15; Smooth Surfaces Detail, 0.5 or 1 µm) on UPEC channel to calculate total extracellular volume in the field of view.

## Isolation and labeling of human neutrophils from fresh human blood

Fresh human blood was procured from voluntary donations by anonymised healthy donors via the Transfusion Interregionale CRS network based in Bern, Switzerland. Donors were made aware that their donation could be used for biomedical research. No personal information about the donors was available to the end users, therefore the consent to publish could not obtained. Approval for this use of blood was provided by the Transfusion Interregionale CRS network (project number P_257). Human neutrophils were isolated via negative depletion method from human blood with MACSxpress Whole Blood Neutrophil Isolation Kit (Miltenyi Biotec), following the manufacturer's instructions. Isolation was performed without the use of a density-based centrifugation method. Isolated human neutrophils were then incubated with a 1 µM solution CellTracker Deep Red in a serum free RPMI phenol red free medium for 30 min in a cell culture incubator maintained at 37°C and 5% $CO_2$. Labelled human neutrophils were then washed with 10 ml of 20% FBS in RPMI phenol-red-free medium followed by centrifugation twice to remove the unbound dye. The human neutrophils were suspended in Lonza EBM2 medium at a cell density of 2 million cells/ml. In some instances, labeled human neutrophils were passed through a filter with 40 µm pores to remove neutrophil clusters that may have formed during the isolation process.

## Scanning electron microscopy of human bladder-chips

UPEC within infected human bladder-chips were allowed to proliferate for 6 hr (ca. 13.5 hr from start of UPEC infection) until the end of the first IBC growth cycle. After 6 hr, bladder-chips were fixed at room temperature for 1 hr with a mix of 1% glutaraldehyde and 2% paraformaldehyde in 0.1 M phosphate buffer (pH 7.4). The fixed bladder-chips were kept in the fixative overnight (at 4°C). Post

overnight fixation, bladder-chips were cut open from the apical channel side. A scalpel was used to cut approximately in the middle of the apical channel side (height=1mm) to expose the bladder epithelial cells. The fixed chips were further fixed for 30 min in 1% osmium tetroxide in 0.1 M cacodylate buffer followed by washing with the distilled water. Next, the bladder-chips were dehydrated in a graded alcohol series and dried by passing them through the supercritical point of carbon dioxide (CPD300, Leica Microsystems). Finally, the bladder-chip was attached to an adhesive conductive surface followed by coating with a 3–4 nm thick layer of gold palladium metal (Quorum Q Plus, Quorum Technologies). Images of the cells were captured using a field emission scanning electron microscope (Merlin, Zeiss NTS).

Uninfected bladder-chip controls were fixed at the same time point (ca. 13.5 hr from start of the experiment). For the case of NETs formed on epithelial layer, an uninfected human bladder-chip was fixed 2 hr after the introduction of neutrophils into the vascular channel of the chip.

### Preparation for transmission electron microscopy (TEM)

UPEC within infected human bladder-chips were allowed to proliferate for 6 hr (ca. 13.5 hr from start of UPEC infection) during the first IBC growth cycle. After 6 hr, bladder-chips were fixed at room temperature for 1 hr with a mix of 1% glutaraldehyde and 2% paraformaldehyde in 0.1 M phosphate buffer (pH 7.4). The fixed bladder-chips were kept in the fixative overnight (at 4°C). Post overnight fixation, bladder-chips were cut open from the apical channel side. A scalpel was used to cut approximately in the middle of the apical channel side (height-1mm) to expose the bladder epithelial cells. The bladder-chips were then washed in cacodylate buffer (0.1M, pH 7.4), postfixed for 40 min in 1.0% osmium tetroxide with 1.5% potassium ferrocyanide, and then fixed again with 1.0% osmium tetroxide alone. The bladder-chips were finally stained for 30 min in 1% uranyl acetate in water before being dehydrated through increasing concentrations of alcohol and then embedded in Durcupan ACM (Fluka, Switzerland) resin. The bladder-chips were then placed in Petri dishes so that approximately 1 mm of resin remained above the cells, and the dish than left in an oven at 65°C for 24 hr. Regions on interest, and corresponding to structures imaged with light microscopy were trimmed from the rest of the device, once the resin had hardened, and thin, 50 nm-thick sections were cut with a diamond knife, and collected onto single-slot copper grids with a pioloform support film. These were contrasted with lead citrate and uranyl acetate, and images taken with a transmission electron microscope at 80 kV (Tecnai Spirit, FEI Company with Eagle CCD camera).

## Acknowledgements

VVT gratefully acknowledges support by a Human Frontier Science Program (HFSP) Long-Term Fellowship (LT000231/2016 L) and a European Molecular Biology Organization (EMBO) Long-Term Fellowship (921-2015). This research was supported by a grant awarded to JDM by the Swiss National Science Foundation (SNSF) (Project Funding, 310030B_176397) and the National Centre of Competence in Research AntiResist (51NF40_180541), funded by the SNSF. The authors thank the entire team of EPFL Bioimaging and Optics Core Facility for their assistance in confocal live cell imaging and post analysis in Bitplane Imaris. The authors also acknowledge Marie Croisier at the EPFL Biological Electron Microscopy Facility for help in optimizing the protocol for TEM of infected bladder-chip. The authors thank and credit BioRender.com for the illustrations and schematics used in this manuscript.

## Additional information

### Funding

| Funder | Grant reference number | Author |
| --- | --- | --- |
| Schweizerischer Nationalfonds zur Förderung der Wissenschaftlichen Forschung | 310030B_176397 | John D McKinney |
| Schweizerischer Nationalfonds zur Förderung der Wissenschaftlichen Forschung | 51NF40_180541 | John D McKinney |

| Human Frontier Science Program | LT000231/2016-L | Vivek V Thacker |
| European Molecular Biology Organization | 921-2015 | Vivek V Thacker |

The funders had no role in study design, data collection and interpretation, or the decision to submit the work for publication.

### Author contributions

Kunal Sharma, Conceptualization, Resources, Data curation, Software, Formal analysis, Investigation, Visualization, Methodology, Writing - original draft, Project administration, Writing - review and editing; Neeraj Dhar, Conceptualization, Resources, Supervision, Validation, Project administration, Writing - review and editing; Vivek V Thacker, Conceptualization, Resources, Supervision, Funding acquisition, Validation, Methodology, Writing - original draft, Project administration, Writing - review and editing; Thomas M Simonet, Resources, Writing - review and editing; Francois Signorino-Gelo, Resources; Graham W Knott, Resources, Validation, Methodology, Writing - review and editing; John D McKinney, Conceptualization, Supervision, Funding acquisition, Project administration, Writing - review and editing

### Author ORCIDs

Kunal Sharma (ID) https://orcid.org/0000-0001-8086-3436
Neeraj Dhar (ID) https://orcid.org/0000-0002-5887-8137
Vivek V Thacker (ID) https://orcid.org/0000-0002-1681-627X
Thomas M Simonet (ID) https://orcid.org/0000-0003-3259-4942
Francois Signorino-Gelo (ID) https://orcid.org/0000-0001-6779-5678
John D McKinney (ID) https://orcid.org/0000-0002-0557-3479

### Ethics

Human subjects: Fresh human blood was procured from anonymised donors via the Transfusion Interregionale CRS network based in Bern, Switzerland. Approval for this project was provided by the same organisation under project number P_257.

### Decision letter and Author response

Decision letter https://doi.org/10.7554/eLife.66481.sa1
Author response https://doi.org/10.7554/eLife.66481.sa2

# Additional files

### Supplementary files

• Transparent reporting form

### Data availability

Data generated in this study has been uploaded to the EPFL community page at Zenodo and is available at the following https://doi.org/10.5281/zenodo.5028262.

The following dataset was generated:

| Author(s) | Year | Dataset title | Dataset URL | Database and Identifier |
| --- | --- | --- | --- | --- |
| Sharma K, Thacker VV | 2021 | Dataset - Dynamic persistence of intracellular bacterial communities of uropathogenic Escherichia coli in a human bladder-chip model of urinary tract infections | https://zenodo.org/record/5028262 | Zenodo, 10.5281/zenodo.5028262 |

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
