## [Decision Letter]

**Acceptance summary:**

Reviewers value the development and characterization of a bladder-on-chip infection model for recapitulating the multiples factors involved in UPEC driven UTIs. Notably, it consists of human bladder epithelial cells (cell line or primary), bladder microvascular endothelial cells, neutrophils and urine that are also subjected to mechanical changes mimicking those occurring during bladder filling and micturition. This model is a lot more complex than in vitro tissue culture models and more amenable to analysis such as imaging than animal models and therefore constitute a distinct advance for in vitro modeling of UTI that has potential to reveal key aspects of UTIs and reasons for the difficulty to clear these infections with antibiotics.

**Decision letter after peer review:**

Thank you for submitting your article "Dynamic persistence of UPEC intracellular bacterial communities in a human bladder-chip model of urinary tract infection" for consideration by *eLife*. Your article has been reviewed by 3 peer reviewers, one of whom is a member of our Board of Reviewing Editors, and the evaluation has been overseen by Gisela Storz as the Senior Editor. The following individual involved in review of your submission has agreed to reveal their identity: Milica Radisic (Reviewer #2).

Essential Revisions:

A few points need addressing to strengthen the study:

– A limitation of the work represents the use of an epithelial cell line. Is it possible to repeat some aspects of this work with a primary epithelial cells for a greater relevance?

– Please repeat the experiment with at least another AB that can reach the intracellular bacteria, since ampicillin will not.

– Reword the paper to avoid overinterpretation of the data in paying particular attention to avoid the following:

1. The immediate interpretation of all intracellular structures as IBCs. IBCs were first described in mice and IBCs usually refer to a specific structure that is formed (at least) in mice and humans in vivo. Similar structures previously observed in vitro are rightfully carefully described such as "structures resembling IBCs". Please use similar care here.

2. The immediate interpretation of all data in Figure 2 as neutrophil swarms and NETs.

3. The claim that a 10% linear change in dimension is "physiologically relevant" and "a significant proportion" of that seen in vivo.

There are occasional sentences where the appropriate care is taken but this is interspersed with writing that is assuming the point is already proven (for example, see lines 286 [appropriate] and 287-289 [not]; and 471-476 [appropriate], 477-479 [not], 481-483 [appropriate]).

---

## [Author Response]

Essential Revisions (for the authors):A few points need addressing to strengthen the study:- A limitation of the work represents the use of an epithelial cell line. Is it possible to repeat some aspects of this work with a primary epithelial cells for a greater relevance?

We agree with the Reviewers in that a primary epithelial cell model would be a closer mimic to the in vivo physiology. However, a complete set of experiments with primary cells is beyond the scope of the manuscript. We have therefore repeated certain key experiments with primary cells pertaining to the formation of IBCs and the subsequent treatment with fosfomycin (related to point 2). Owing to the constraints imposed by the COVID-19 pandemic, these experiments were limited to snapshots taken at critical timepoints in the antibiotic treatment cycle. In the revised manuscript, we provide clear evidence of the formation of numerous IBCs in bladder-chips reconstituted with primary cells (Figure 3 —figure supplement 1 G-I), as well as snapshot evidence of regrowth despite two rounds of fosfomycin treatment (Figure 4 —figure supplement 1 A, B). These results reinforce the strength of our data with the cell line and ampicillin.

- Please repeat the experiment with at least another AB that can reach the intracellular bacteria, since ampicillin will not.

We disagree somewhat with the reviewers in their categorical statement that ampicillin does not penetrate host cells; there are studies that show host cell penetration and activity against a range of pathogens (e.g., Lemaire et al., PMID: 15860552, Carryn et al., PMID: 12069960, and Prabhakaran et al., PMID: 10595573). Ampicillin was prescribed for urinary tract infections prior to the widespread development of antibiotic resistance (Bonadio *et al.*, PMID: 11713400). Our choice for the use of ampicillin was also motivated by the fact that this antibiotic has been used extensively in laboratory experiments with bacteria under axenic conditions. However, we do agree that current first-line therapeutics such as fosfomycin have an increased cell penetrability and a different bacteriocidal mechanism. We have therefore repeated the antibiotic treatment experiments with fosfomycin for both primary cells (snapshots only) and the 5637-cell line (lower resolution imaging to provide higher throughput) and included this data in the revised manuscript. We observe IBCs both in 5637 cells (new Figure 3 —figure supplement 3 E, F) as well as primary human bladder epithelial cells (new Figure 3 —figure supplement 3 G-I). Although fosfomycin reduces IBC volume by a significantly larger fraction than ampicillin (new Figure 4I, 4K) we nevertheless observe regrowth from IBCs through two rounds of treatment (new Figure 4 —figure supplement 1). This data significantly strengthens our overall conclusion that IBCs may be a protective niche against antibiotic treatment for a prolonged period of time.

- Reword the paper to avoid overinterpretation of the data in paying particular attention to avoid the following:1. The immediate interpretation of all intracellular structures as IBCs. IBCs were first described in mice and IBCs usually refer to a specific structure that is formed (at least) in mice and humans in vivo. Similar structures previously observed in vitro are rightfully carefully described such as "structures resembling IBCs". Please use similar care here.

We agree with the reviewer that although certain aspects of the structures we observe conform to established descriptions of IBCs, they may differ from IBCs formed in infected animals in other respects. In the Revised Manuscript between Lines 331 – 358 we have altered the flow of the narrative to first describe the morphological characteristics of these structures before assigning them as IBCs. Furthermore, we have therefore inserted the following sentence at Lines 347-349:

“Hereafter, we shall refer to these IBC-like structures as IBCs for simplicity, acknowledging that these structures may differ from IBCs reported in infected animals.”

2. The immediate interpretation of all data in Figure 2 as neutrophil swarms and NETs.

We agree with the reviewer that our interpretation in the text of the data in Figure 2 was premature. In the Revised Manuscript, we have edited the text between Lines 271 – 310 at numerous places to describe the evidence from the immunostaining experiments and SEM images first before classifying the bundles that form between neutrophils as NETs.

3. The claim that a 10% linear change in dimension is "physiologically relevant" and "a significant proportion" of that seen in vivo.

We have edited the Revised Manuscript to remove the two descriptors. In the revised manuscript, at Line 171-172, the bladder-chip model is described as:

“with an applied strain that partly mimics the physiology of bladder filling and voiding cycles”

Furthermore, in the Revised Manuscript at Lines 163-165, we have clarified that the applied strain of 10% is a small fraction of the maximum applied strain in the bladder.

“We therefore limited the linear strain applied to a maximum of 10%, which is of the same order of magnitude, albeit a small fraction of the typical strain experienced by the bladder tissue in vivo.”

There are occasional sentences where the appropriate care is taken but this is interspersed with writing that is assuming the point is already proven (for example, see lines 286 [appropriate] and 287-289 [not]; and 471-476 [appropriate], 477-479 [not], 481-483 [appropriate]).

We apologies for the error on our part. We thank the reviewers for pointing out these specific instances and we have edited these sentences as part of our response to the earlier points so that the lines of evidence are listed first before the conclusion is drawn. The instances pointed out by the Reviewers at lines 477-489 and 481-483 corresponded to text in the Discussion which, in the original manuscript, partly repeated evidence already presented in the Results section. For ease of reading, in the revised manuscript we have edited these lines to avoid referring to the primary data. We hope that with the revised text in the Results section, the reviewers and readers are sufficiently convinced about the strength of our findings that we do not need to present the evidence again in the Discussion.